# GenAR: Next-Scale Autoregressive Generation for Spatial Gene Expression Prediction

## Abstract

Spatial Transcriptomics (ST) offers spatially resolved gene expression but remains costly. Predicting expression directly from widely available Hematoxylin and Eosin (H&E) stained images presents a cost-effective alternative. However, most computational approaches (i) predict each gene independently, overlooking co-expression structure, and (ii) cast the task as continuous regression despite expression being discrete counts. This mismatch can yield biologically implausible outputs and complicate downstream analyses. We introduce **GenAR**, a multi-scale autoregressive framework that refines predictions from coarse to fine. GenAR (a) clusters genes into hierarchical groups to expose cross-gene dependencies, (b) models expression as discrete token generation over a fixed vocabulary of integer count tokens to directly predict raw counts, and (c) conditions decoding on fused histological and spatial embeddings. From an information-theoretic view, the discrete formulation operates directly on the physical count scale, and the coarse-to-fine factorization aligns with a principled conditional decomposition. Extensive experimental results on four ST datasets across different tissue types demonstrate that GenAR achieves state-of-the-art performance, offering potential implications for precision medicine and cost-effective molecular profiling. Code will be publicly available.

## 1 Introduction

Spatial Transcriptomics (ST) has emerged as a transformative technology, enabling measurement of gene expression while preserving the spatial organization of cells within tissue samples Jain & Eadon (2024); Rao et al. (2021b); Xiao & Yu (2021). Unlike traditional bulk RNA sequencing, which averages gene expression across entire tissue samples and discards spatial context, ST maintains the spatial relationships among cells and their molecular profiles. This spatially resolved approach reveals how gene expression patterns vary across different tissue regions, providing molecular insights that complement conventional morphological assessment such as Hematoxylin and Eosin (H&E) staining Ilse et al. (2018); Yang et al. (2024); Chen et al. (2025b); Xu & Chen (2023). The impact of ST technology extends across multiple biomedical domains, such as cancer research, where it identifies spatially distinct tumor subregions Bera et al. (2019).

However, ST technology faces significant practical barriers that limit its widespread adoption. Current ST protocols require specialized laboratory equipment, extensive technical expertise, and considerable time investment. Per-sample costs often range from hundreds to thousands of dollars, making large-scale studies financially challenging Rao et al. (2021a). These constraints have resulted in relatively small ST datasets, whereas H&E images are abundant and inexpensive to obtain. This scarcity of ST data further reduces the practical utility of this technology and hinders comprehensive spatial studies across diverse tissue types and disease conditions.

To address this challenge, several computational methods have been proposed for predicting spatial gene expression directly from histopathological images. Early studies such as ST-Net He et al. (2020) and Hist2ST Zeng et al. (2022) established the basic framework for linking morphological features to molecular profiles. Subsequent work includes BLEEP Xie et al. (2023), which implemented bi-modal embedding with contrastive learning, TRIPLEX Chung et al. (2024), which integrates multi-resolution feature, and M2OST Wang et al. (2025) with multimodal and multi-scale

strategies. More recently, STEM Zhu et al. (2025) attempted to solve this problem using conditional diffusion models from a generative modeling perspective.

Despite these significant advances, existing methods share several limitations: First, many methods independently predict the expression of each gene, underutilizing cross-gene dependencies. Genes rarely function in isolation but rather operate in concert through regulatory networks, signaling pathways, and co-expression modules Barabasi & Oltvai (2004). Treating genes as independent targets can therefore miss biologically meaningful interactions.

Second, existing methods face challenges in maintaining biological interpretability due to their modeling of gene expression as continuous regression tasks. Gene expression is recorded as nonnegative integer counts that approximate the number of mRNA molecules per spot or cell, typically ranging from zero to several thousand. These raw counts carry important meaning for biological applications such as differential expression and pathway enrichment Love et al. (2014). However, current approaches apply a $\log$ transformation to gene expression data, converting them into continuous floating-point numbers (typically 0-15) for prediction. This transformation departs from the discrete count scale used in biological analyses and may lead to predictions that cannot be directly interpreted in terms of molecule counts.

To address these challenges, we propose GenAR (**Gen**e expression prediction via next-scale **A**uto**R**egressive), a progressive multi-scale autoregressive framework that overcomes the limitations of existing approaches. Specifically, GenAR tackles the aforementioned problems as follows: First, rather than predicting genes independently, we cluster genes into coarse-to-fine groups and perform sequential prediction across scales; each scale conditions on all prior predictions to encode cross-gene structure and progressively refine estimates. Second, our framework directly predicts raw gene expression counts, keeping biological meaning intact and allowing direct use in biological analyses. Third, we cast prediction as discrete token generation rather than continuous regression—an information-theoretic, entropy-preserving view that operates directly on the count scale and avoids relying on log-transformed surrogates, while aligning with a conditional probability decomposition.

Our main contributions can be summarized as follows:

- We propose a progressive multi-scale autoregressive framework for spatial gene expression prediction that decomposes the prediction task into sequential scales from coarse to fine granularity.

- We develop a discrete token generation approach that directly predicts raw gene expression counts through a fixed count-token vocabulary without a learned latent codebook, preserving biological interpretability and enabling direct use in downstream analyses.

- GenAR demonstrates state-of-the-art performance on four spatial transcriptomics datasets, outperforming existing methods across standard evaluation metrics.

## 2 RELATED WORK

### 2.1 GENE EXPRESSION PREDICTION

Initial work in this field focused on connecting tissue morphology with molecular profiles. ST-Net He et al. (2020) applied the DenseNet architecture to extract features from H&E stained images for gene expression prediction. Hist2ST Zeng et al. (2022) advanced this approach by combining convolutional networks, Transformers, and graph neural networks to better capture complex spatial relationships and cellular interactions in tissue samples, demonstrating the importance of modeling spatial context in gene expression prediction. Histogene Pang et al. (2021) brought the Vision Transformer architecture to this domain, leveraging self-attention mechanisms to capture long-range dependencies in histopathological images that traditional convolutional approaches might miss.

As the field evolved, specialized methods such as BLEEP Xie et al. (2023) introduced bi-modal embeddings and contrastive learning, aligning histopathological and gene expression data more effectively. EGN Yang et al. (2023) utilized exemplar-guided networks for efficient spatial transcriptomics analysis by learning from representative samples, reducing computational overhead while maintaining prediction accuracy.

Recent work has explored multimodal and multi-scale strategies to further improve performance. TRIPLEX Chung et al. (2024) designed a multi-resolution framework with three specialized encoders to capture local patch features, spatial context, and tissue-level patterns at different scales. UMPIRE Han et al. (2024) employed contrastive learning to align image and gene expression representations using large-scale paired datasets, showing that scaling up training data significantly improves model generalization. M2OST Wang et al. (2025) enhanced prediction accuracy by integrating multimodal information and multi-scale feature representations, STEM Zhu et al. (2025) attempted to solve this problem from a generative modeling perspective using conditional diffusion models, treating gene expression prediction as a generation task conditioned on histological features.

## 2.2 Next-Scale Autoregressive Generation

Autoregressive generation has achieved remarkable success in natural language processing and computer vision. Early visual autoregressive approaches treated images as sequences of pixels Van den Oord et al. (2016), but suffered from computational inefficiency. Vector Quantized Variational AutoEncoder (VQ-VAE) Van Den Oord et al. (2017) addressed this by representing images as discrete token sequences through quantization, establishing a two-stage paradigm of discretization followed by autoregressive prediction.

Recently, VAR Tian et al. (2024) proposed next-scale prediction, generating images progressively from coarse to fine scales rather than sequential token-by-token generation. VAR has inspired extensions across diverse applications Ma et al. (2024); Qu et al. (2025); Chen et al. (2025a), establishing next-scale autoregressive generation as a promising paradigm. These methods follow a two-stage paradigm where a VQ-VAE first discretizes continuous visual data into codebook tokens, which are then predicted by an autoregressive model. This approach is necessitated by the continuous nature of visual data, requiring explicit discretization to enable autoregressive modeling.

However, gene expression prediction presents a unique opportunity to leverage the inherently discrete nature of the data. Since gene expression counts are naturally discrete integer values, we can directly apply autoregressive modeling without requiring a learned VQ-VAE–style codebook. This yields an end-to-end pipeline that uses a non-learned quantization of counts (each integer count is mapped to a dedicated token) rather than a latent codebook, avoiding an extra encode–decode reconstruction stage and its potential information loss.

## 3 Methodology

**Problem Formulation.** Let $\mathcal{G} = \{1, \ldots, n\}$ index the genes. For each spatial location $u$ (spot), we observe an H&E image patch $I_u \in \mathbb{R}^{\mathcal{H} \times \mathcal{W} \times 3}$ and its coordinates $S_u \in \mathbb{R}^2$. The target is a vector of nonnegative integer counts $\mathbf{y}_u \in \mathbb{N}_0^n$, where $y_{u,g}$ denotes the expression count of gene $g \in \mathcal{G}$ at location $u$ Anders & Huber (2010). Given a dataset $\mathcal{D} = \{(I_u, S_u, \mathbf{y}_u)\}_{u=1}^N$, the goal is to learn a mapping that predicts gene expression counts $\hat{\mathbf{y}}_u = \text{GenAR}_\theta(I_u, S_u)$. Training minimizes the expected loss over $\mathcal{D}$:

$$\theta^\star = \arg\min_\theta \ \frac{1}{N} \sum_{u=1}^N \mathcal{L}(\mathbf{y}_u, \hat{\mathbf{y}}_u) \tag{1}$$

where $\mathcal{L}$ is the multi-scale loss function defined in Section 3.

**Overview of GenAR.** An overview of GenAR is shown in Figure 1. Genes are reordered into hierarchical clusters based on spatial expression patterns, progressing from major gene groups to smaller nested subgroups. Given an H&E patch $I_u$ and its coordinates $S_u$, we first extract histopathological features using a pre-trained foundation model Chen et al. (2024). We then incorporate spatial context by applying a sinusoidal positional encoding to $S_u$. Both modalities are processed through a fusion module to obtain a final histological embedding $H \in \mathbb{R}^{768}$.

GenAR employs a progressive multi-scale autoregressive framework to capture cross-gene structure that independent regression baselines often underutilize. We define $K$ scales with hierarchical gene groups $\{\mathcal{C}^{(1)}, \ldots, \mathcal{C}^{(K)}\}$ from coarse to fine, starting from a single global group and refining to smaller groups and finally individual genes. At each scale $k$, the model predicts grouped expressions $\mathbf{y}_u^{(k)}$ conditioned on all previously generated coarser outputs $\mathbf{y}_u^{(<k)}$.

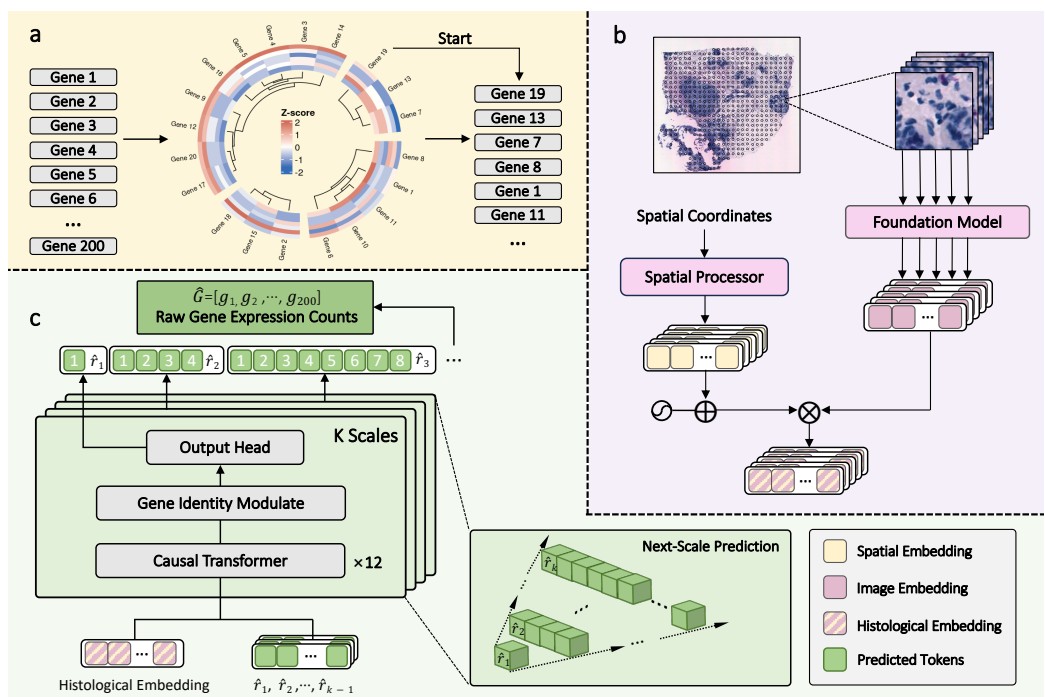

Figure 1: Overall architecture of GenAR. (a) Genes are clustered into hierarchical groups from coarse to fine granularity. (b) Image and spatial features are fused to generate histological embeddings. (c) Multi-scale autoregressive generation progressively refines predictions across scales.

At each scale, we represent gene expression counts as discrete tokens and map them to dense vectors via a learned embedding layer. The resulting sequence is processed by a causal Transformer decoder that is conditioned on the histological embedding $H$ through adaptive layer normalization (AdaLN) Dhariwal & Nichol (2021). We further apply feature-wise linear modulation, where gene-identity embeddings produce scale and shift parameters to inject gene-specific inductive bias into the model. Finally, the decoder outputs token logits for the current scale, which are then converted into integer expression counts.

**Gene Clustering and Histological Embeddings.** We cluster genes based on their spatial expression patterns in the training set. Using $k$-means on $Z$-score normalized expression profiles, we first group the 200 genes into 4 major clusters, then subdivide each cluster into smaller groups of approximately 12 genes.

The fusion module applies layer normalization to histopathological features $\phi(I_u) \in \mathbb{R}^{1024}$, followed by two linear layers with GELU activation and dropout regularization. Spatial information $S_u \in \mathbb{R}^2$ undergoes sinusoidal positional encoding to capture spatial relationships, followed by linear projection and normalization. The processed features are concatenated and projected to the final histological embeddings dimension $H \in \mathbb{R}^{768}$.

Gene expression counts are mapped to dense representations through a learned embedding layer $E_{\text{gene}} \in \mathbb{R}^{\text{vocab\_size} \times 768}$. Considering that gene expression counts typically range from 0 to several thousand, we adopt a fixed-size vocabulary to cover this range. Gene identity embeddings $E_{\text{identity}} \in \mathbb{R}^{n \times 768}$ capture the characteristics and functional properties of each gene. Gene modulation is achieved through feature-wise linear modulation, where gene identity embeddings are transformed to generate scaling and shift parameters that modulate the hidden representations.

Appendix A.6 demonstrates that replacing spatial pattern–based groups with random groupings substantially degrades performance. Appendix A.7 reveal that many learned groups align with known pathways and cell-type modules, indicating that the hierarchy is both empirically beneficial and biologically meaningful.

---

**Algorithm 1** GenAR Training Process

---

**Require:** Histology patches $I_u$, Spatial coordinates $S_u$, Ground-truth counts $\mathbf{y}$
**Ensure:** Final training loss $L_{\text{final}}$
 1: $H \leftarrow \text{ConditionProcessor}(I_u, S_u)$ ▷ Fuse multi-modal context
 2: $\mathcal{T} \leftarrow \text{CreateHierarchicalTargets}(\mathbf{y})$ ▷ Prepare multi-scale ground truth
 3: $\mathcal{E}_{\text{outputs}} \leftarrow \emptyset, \quad L_{\text{total}} \leftarrow 0$
 4: **for** each scale $k \in \{1, \ldots, K\}$ with dimension $d_k$ **do**
 5:     **if** $k = 1$ **then**
 6:         $X_{\text{context}} \leftarrow [\text{START\_TOKEN}]$
 7:     **else**
 8:         $G_{\text{context}} \leftarrow \text{GetTokensFromTargets}(\mathcal{T}_{<k})$ ▷ Teacher forcing with cumulative history
 9:         $X_{\text{context}} \leftarrow \text{Concat}([\text{START\_TOKEN}], \text{GeneEmbed}(G_{\text{context}}))$
10:     **end if**
11:     $X_{\text{init}} \leftarrow \text{GeneUpsampling}(\mathcal{E}_{\text{outputs}}, k)$ ▷ Initialize current scale's targets
12:     $X \leftarrow \text{Concat}(X_{\text{context}}, X_{\text{init}}) + \text{PosEmbed}(k) + \text{ScaleEmbed}(k)$
13:     $X_{\text{hidden}} \leftarrow \text{Transformer}(X, H, \text{CausalMask})$
14:     $\text{Logits} \leftarrow \text{OutputHead}(\text{FiLM}(\text{SliceLastTokens}(X_{\text{hidden}}, d_k), \text{GeneIdentity}(k)))$
15:     **if** $k < K$ **then**
16:         $L_k \leftarrow \text{SoftKLLoss}(\text{Logits}, \mathcal{T}_k)$ ▷ Group-level soft supervision
17:     **else**
18:         $L_k \leftarrow \text{GaussianNLL}(\text{CountHead}(\text{Logits}), \mathcal{T}_k; \sigma^2 = \alpha \, \text{CountHead}(\text{Logits}) + \beta)$ ▷
    Count-level heteroscedastic loss
19:     **end if**
20:     $L_{\text{total}} \leftarrow L_{\text{total}} + L_k$
21:     $\mathcal{E}_{\text{outputs}}.\text{Append}(\text{GeneEmbed}(\text{ArgMax}(\text{Logits})))$ ▷ Update state for next scale
22: **end for**
23: $L_{\text{final}} \leftarrow L_{\text{total}}/K$
24: **return** $L_{\text{final}}$

---

**Progressive Multi-Scale Generation.** The autoregressive generation process can be formalized as:

$$p(\mathbf{y} \mid H) = \prod_{k=1}^{K} p(\mathbf{y}^{(k)} \mid H, \mathbf{y}^{(<k)}) \tag{2}$$

where $\mathbf{y}^{(k)}$ denotes expressions at scale $k$, $\mathbf{y}^{(<k)}$ denotes all previous-scale outputs, and $H$ represents the histological embeddings derived from histopathological patches and spatial information.

We design $K$ sequential scales to capture gene expression relationships at different granularities, i.e., a structured conditional factorization from global to gene-level interactions. At each scale $k$, genes are divided into $d_k$ groups, where each group contains consecutive genes from the cluster gene ordering. The number of groups increases progressively across scales: the first scale uses a single group representing global transcriptional activity across all genes, intermediate scales gradually increase the number of groups to capture finer-grained patterns, and the final scale contains individual genes for precise prediction. We design this hierarchical decomposition to allow GenAR to establish dependencies between genes at different levels of granularity, moving from global transcriptional context to specific gene interactions.

The autoregressive property applies across scales, where predictions at each scale are conditioned on all previously generated coarser-grained information. At each scale, gene expression values are tokenized and embedded, then processed through a causal Transformer architecture conditioned on the embeddings $H$. Our approach does not rely on a learned VQ-VAE–style codebook: it directly treats each integer count as a dedicated token and predicts these tokens autoregressively, without any separate vector-quantization or codebook-learning stages. This non-learned count tokenization eliminates potential reconstruction loss from an encoder–decoder bottleneck and keeps the overall pipeline fully end-to-end.

As illustrated in Figure 2, our framework operates differently during training and inference phases. During training (left panel), the model learns to predict tokens at each scale using ground-truth information from previous scales. The process begins with a start token, followed by ground-truth

Figure 2: Progressive multi-scale generation process, illustrating sequence construction and upsampling initialization during training and inference phases.

tokens from completed scales, and interpolated tokens that provide initialization for the current scale prediction.

During inference (right panel), the model generates predictions autoregressively across scales. At each scale $k$, the input sequence is constructed as:

$$X_k = [\text{start\_token}, \hat{\mathbf{y}}^{(1)}, \ldots, \hat{\mathbf{y}}^{(k-1)}, \text{interpolated\_tokens}_k] \tag{3}$$

where $\hat{\mathbf{y}}^{(j)}$ represents previously generated tokens from scale $j$. The interpolated tokens are obtained through upsampling operations from the previous scale's embeddings, providing contextual initialization for current scale generation.

**Multi-Scale Loss Function.** Under the hierarchical factorization, the negative log-likelihood decomposes across scales:

$$\mathbb{E}_q\big[-\log p_\theta(\mathbf{y} \mid H)\big] = \sum_{k=1}^{K} \mathbb{E}_q\big[-\log p_\theta(\mathbf{y}^{(k)} \mid H, \mathbf{y}^{(<k)})\big] = \sum_{k=1}^{K} \text{KL}\big(q^{(k)} \,\|\, p_\theta^{(k)}\big) + \text{const}, \tag{4}$$

where $q^{(k)}$ is the target distribution at scale $k$ (group levels use temperature-smoothed targets derived from adaptive pooling with a fixed temperature $\tau = 1.0$; the final level reduces to a sharp target over counts) and $p_\theta^{(k)}$ is the model distribution.

At intermediate scales, we supervise grouped targets obtained by $\mathbf{y}^{(k)} = \text{AdaptiveAvgPool1d}(\mathbf{y}, d_k)$ for $k < K$ and convert them to soft distributions via temperature smoothing, optimizing $\text{KL}(q^{(k)}\|p_\theta^{(k)})$. At the final scale, we use a count-level likelihood with expression-dependent variance $\sigma^2 = \alpha\mu + \beta$ (equivalently a KL to a Gaussian family up to a constant) to capture heteroscedasticity while preserving count semantics; here $\text{CountHead}(\cdot)$ maps final-scale logits to the mean $\hat{\mu}$ used in the Gaussian NLL. The overall objective averages losses across scales:

$$\mathcal{L}_{\text{total}} = \frac{1}{K} \sum_{k=1}^{K} \mathcal{L}_k. \tag{5}$$

In all experiments we set the per-scale weights $\lambda_k$ to 1 and simply average the per-scale losses, i.e., $\mathcal{L}_{\text{total}} = \frac{1}{K} \sum_k \mathcal{L}_k$; Appendix A.6 shows that heavily down-weighting or dropping the intermediate scales leads to large performance drops, confirming the importance of strong supervision at all levels.

## 4 EXPERIMENTS

### 4.1 DATASETS

We conducted experiments on four different spatial transcriptomics datasets selected from the HEST-1k database Jaume et al. (2024), spanning multiple tissue types and disease states.

**HER2ST dataset** Andersson et al. (2021) contains breast cancer tissue slides with spatial spots of 100 $\mu$m diameter. This dataset consists of multiple pathology images with a total of 13,594 spots, each containing gene expression profiles. The tissue samples include normal breast tissue and

cancerous regions. In our experiments, we used the SPA148 slide as the test set, with the remaining slides used for training.

**Human Prostate Cancer (PRAD) Visium dataset** Erickson et al. (2022) contains 23 prostate cancer tissue slides sequenced using the 10x Genomics Visium platform. The spatial spots have a size of 55 $\mu$m, with the number of spots per slide ranging from 1,418 to 4,079. We used the MEND145 slide as the test set, with the remaining slides used for training.

**Kidney Visium dataset** Lake et al. (2023) contains 23 kidney tissue slides from samples representing three pathological states: healthy controls, chronic kidney disease, and acute kidney injury. The data was acquired using Visium technology with spatial spots of 55 $\mu$m size, and the number of spots per slide ranges from 315 to 4,159. The samples cover both cortical and medullary anatomical regions of the kidney. We used the NCBI697 slide as the test set, with the remaining slides used for training.

**Healthy Mouse Brain dataset** Vicari et al. (2024) contains 14 Visium samples from healthy adult mouse brain tissue. Each slide contains 2,675 to 3,617 spatial spots with a spot size of 55 $\mu$m. We selected the NCBI667 slide as the test set, with the remaining slides used for training.

## 4.2 DATA PREPROCESSING AND EVALUATION METRICS

**Data Preprocessing.** For all four datasets, we applied a consistent preprocessing pipeline Zhu et al. (2025). We selected the top 200 genes from the intersection of highly expressed and highly variable genes for evaluation. For image processing, we used a patch size of $224 \times 224$ pixels for all datasets, with each patch corresponding to one spatial spot. Our model extracts histopathological image features using UNI Chen et al. (2024). For gene expression count processing, while other baseline models applied $\log_2$ transformation, following Jaume et al. (2024), our model directly predicts raw gene expression counts without $\log_2$ transformation during training. To ensure consistent evaluation with other models, we then applied $\log_2$ transformation to our model outputs after prediction for metric calculation. In addition to these log-space metrics, we also evaluate all methods directly on inverse-transformed integer counts (MSE, MAE, Spearman correlation, and a Negative Binomial NLL fit to the predicted mean and empirical variance); these raw-count results are reported in Appendix A.4.

**Evaluation Metrics.** We used multiple evaluation metrics to assess model prediction performance. We used PCC-10, PCC-50, and PCC-200 metrics, representing the average PCC values of the top 10, 50, and 200 genes with the highest Pearson correlation coefficients in the prediction results, respectively. For a single gene $g$, the PCC is calculated as:

$$\text{PCC}_g = \frac{\text{Cov}(Y_g, \hat{Y}_g)}{\sqrt{\text{Var}(Y_g) \cdot \text{Var}(\hat{Y}_g)}} \quad (6)$$

where $Y_g$ and $\hat{Y}_g$ represent the true and predicted expression values for gene $g$, respectively, $\text{Cov}(\cdot)$ denotes covariance, and $\text{Var}(\cdot)$ denotes variance.

Moreover, we calculated Mean Squared Error (MSE) and Mean Absolute Error (MAE) to evaluate the overall prediction accuracy of the model. MSE computes the average squared error between predicted and true values across all spatial spots and genes. MAE computes the average absolute error between predicted and true values across all spatial spots and genes.

**Implementation Details.** Our models are trained with the Adam optimizer Kingma (2014) with a learning rate of 1e-4 and a batch size of 64. For these four datasets, we configured 6 hierarchical scale dimensions (1, 4, 8, 40, 100, 200) for multi-scale feature extraction, targeting the prediction of 200 gene expression counts. The main hyperparameters of our model include model depth, model width, and the number of heads in self-attention mechanisms. Increased model complexity is often accompanied by higher computational resource requirements, our model design balances prediction performance while considering computational efficiency. Experiments were conducted in PyTorch on NVIDIA H100 (80 GB) GPUs. Baselines used the same preprocessing and tuning protocol.

| Method | HER2ST | | | | | PRAD | | | | |
|---|---|---|---|---|---|---|---|---|---|---|
| | PCC-10↑ | PCC-50↑ | PCC-200↑ | MSE↓ | MAE↓ | PCC-10↑ | PCC-50↑ | PCC-200↑ | MSE↓ | MAE↓ |
| BLEEP Xie et al. (2023) | 0.773 | 0.714 | 0.565 | 1.243 | 0.833 | 0.580 | 0.510 | 0.316 | 2.475 | 1.091 |
| M2OST Wang et al. (2025) | 0.810 | 0.759 | 0.660 | 1.151 | 0.820 | 0.602 | 0.551 | 0.442 | 1.290 | 0.862 |
| TRIPLEX Chung et al. (2024) | 0.783 | 0.714 | 0.586 | 1.212 | 0.857 | 0.620 | 0.544 | 0.423 | 1.319 | 0.836 |
| STEM Zhu et al. (2025) | 0.831 | 0.770 | 0.625 | 1.199 | 0.787 | 0.636 | 0.555 | 0.403 | 1.457 | 0.857 |
| **GenAR(Ours)** | **0.842** | **0.784** | **0.663** | **1.082** | **0.745** | **0.702** | **0.650** | **0.512** | **1.191** | **0.771** |

Table 1: Experimental results on HER2ST and PRAD datasets. The best results are highlighted in **bold**. ↑ indicates higher is better, ↓ indicates lower is better.

| Method | Kidney | | | | | Healthy Mouse Brain | | | | |
|---|---|---|---|---|---|---|---|---|---|---|
| | PCC-10↑ | PCC-50↑ | PCC-200↑ | MSE↓ | MAE↓ | PCC-10↑ | PCC-50↑ | PCC-200↑ | MSE↓ | MAE↓ |
| BLEEP Xie et al. (2023) | 0.500 | 0.422 | 0.314 | 1.926 | 0.945 | 0.342 | 0.280 | 0.156 | 1.591 | 0.987 |
| M2OST Wang et al. (2025) | 0.494 | 0.447 | 0.318 | 1.785 | 0.925 | 0.456 | 0.387 | 0.231 | 1.148 | 0.861 |
| TRIPLEX Chung et al. (2024) | 0.542 | 0.469 | 0.336 | 1.732 | 0.887 | 0.501 | 0.445 | 0.312 | 1.157 | 0.822 |
| STEM Zhu et al. (2025) | 0.567 | 0.483 | 0.322 | 1.832 | 0.997 | 0.526 | 0.452 | 0.331 | 1.235 | 0.864 |
| **GenAR(Ours)** | **0.589** | **0.514** | **0.354** | **1.636** | **0.871** | **0.568** | **0.503** | **0.367** | **1.138** | **0.805** |

Table 2: Experimental results on Kidney and Mouse Brain datasets. The best results are highlighted in **bold**. ↑ indicates higher is better, ↓ indicates lower is better.

Results were obtained with fixed seeds. Appendix A.8 reports a FLOPs comparison. Code and configuration files will be released.

## 4.3 EXPERIMENTAL RESULTS

We evaluated the performance of our proposed method on four different spatial transcriptomics datasets and compared it with multiple baseline methods. The experimental results are shown in Table 1 and Table 2. Our method achieves the best performance across all datasets.

As shown in Table 1, on the HER2ST dataset containing breast cancer tissue slides with 100 $\mu$m spatial spots, our method achieves PCC-10, PCC-50, and PCC-200 scores of 0.842, 0.784, and 0.663, outperforming the best baseline method STEM by 1.3%, 1.8%, and 6.1%, respectively. The MSE and MAE are reduced by 9.8% and 5.3% compared to STEM. On the PRAD dataset containing prostate cancer tissue slides with 55 $\mu$m spatial spots, our method demonstrates more significant improvements, achieving PCC-10, PCC-50, and PCC-200 scores of 0.702, 0.650, and 0.512, surpassing STEM by 10.4%, 17.1%, and 27.0%, respectively. The MSE and MAE reductions are 18.3% and 10.0%, respectively.

As shown in Table 2, on the Kidney dataset covering three pathological states with 55 $\mu$m spatial spots, our method achieves PCC-10, PCC-50, and PCC-200 scores of 0.589, 0.514, and 0.354, outperforming STEM by 3.9%, 6.4%, and 9.9%, respectively. The MSE and MAE are reduced by 10.7% and 12.6%. On the Mouse Brain dataset containing healthy adult mouse brain tissue with 55 $\mu$m spatial spots, our method achieves PCC-10, PCC-50, and PCC-200 scores of 0.568, 0.503, and 0.367, surpassing STEM by 8.0%, 11.3%, and 10.9%, respectively. The MSE and MAE reductions are 7.9% and 6.8%.

Cross-dataset analysis shows consistent gains, with larger margins on cancer tissues, indicating that the proposed coarse-to-fine discrete autoregressive formulation is robust across tissue types.

## 4.4 ABLATION STUDY

**Component Ablation.** We perform internal ablation experiments on the PRAD dataset, which contains numerous spatial spots and exhibits high complexity. As shown in Table 3, we separately remove the progressive multi-scale generation framework, gene identity embeddings, and replace the loss function to analyze the contribution of each component.

The experimental results demonstrate that removing the progressive multi-scale generation framework has the most significant impact on model performance, with PCC-10 decreasing from 0.702

| Method | PCC-10↑ | PCC-50↑ | PCC-200↑ | MSE↓ | MAE↓ |
|---|---|---|---|---|---|
| w/o Multi-scale | 0.651 | 0.601 | 0.493 | 1.406 | 0.779 |
| w/o Gene identity | 0.683 | 0.625 | 0.481 | 1.281 | 0.828 |
| w/ Cross-entropy | 0.662 | 0.612 | 0.482 | 1.325 | 0.781 |
| GenAR | **0.702** | **0.650** | **0.512** | **1.191** | **0.771** |

Table 3: Ablation study results on the PRAD dataset. The best results are highlighted in **bold**. ↑ indicates higher is better, ↓ indicates lower is better.

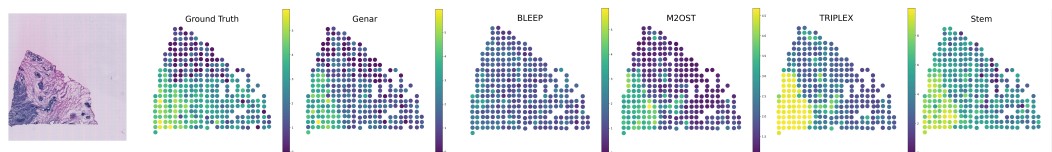

Figure 3: Spatial visualization of SSR4 gene expression prediction on HER2ST SPA148 sample. From left to right: histopathological image, ground truth, and predictions from GenAR, BLEEP, M2OST, TRIPLEX, and STEM. Color scale: low (purple/blue) to high (yellow/green) expression.

to 0.651 and MSE increasing from 1.171 to 1.406. This indicates that the multi-scale autoregressive generation process is crucial for capturing gene expression relationships at different granularities. After removing gene identity embeddings, the PCC-200 metric decreases from 0.532 to 0.481, demonstrating the importance of gene-specific representations for precise prediction. Using cross-entropy loss instead of our designed adaptive Gaussian KL loss and soft-label KL divergence loss results in PCC-10 decreasing from 0.702 to 0.662. We also observe weaker performance in extremely sparse regions (token rarity/gradient sparsity), motivating pathway/ontology-informed grouping without altering the core framework.

**Foundation Model Ablation.** We design a raw gene expression count prediction task and evaluate it across three foundation models and GenAR. We compare ResNet-18 He et al. (2016a), CONCH Huang et al. (2023), and UNI Chen et al. (2024). Our full GenAR model selects the UNI Chen et al. (2024) to extract histological features. For fair comparison, all baselines use identical architectures: input projection to hidden space, two residual blocks with GELU activation, and output heads for discrete gene expression prediction. Table 4 reports results on the PRAD dataset. All three foundation models achieve reasonable performance on this discrete prediction task, while GenAR substantially outperforms them with 33.7% improvement in PCC-200 over the best baseline, demonstrating the effectiveness of our framework design.

Beyond these main ablations, Appendix A.3 compares GenAR to a continuous regression variant with the same multi-scale architecture and shows that the discrete formulation yields substantially higher PCC, especially on low-expression genes, and recovers a much larger fraction of true zeros. Appendix A.5 further stratifies performance by expression level, and Appendix A.6 reports vocabulary utilization statistics, indicating that the discrete count tokenization is effective without collapsing to a tiny subset of bins.

### 4.5 VISUALIZATION ANALYSIS

Figure 3 shows visualization results on the HER2ST dataset using sample SPA148 for gene SSR4 (Signal Sequence Receptor Subunit 4), which encodes an ER membrane receptor associated with cancer progression. The ground truth exhibits distinct spatial heterogeneity with concentrated high-expression regions (yellow-green) and clearly demarcated low-expression areas (purple-blue).

BLEEP and M2OST generate smooth expression maps with limited dynamic range, failing to capture sharp spatial transitions. TRIPLEX and STEM show improved pattern recognition, with STEM preserving better boundaries, though both exhibit oversmoothing in high-expression regions. GenAR produces expression predictions that most closely match ground truth, accurately capturing both high-expression zone localization and expression level transitions.

| Method | PCC-10↑ | PCC-50↑ | PCC-200↑ | MSE↓ | MAE↓ |
|---|---|---|---|---|---|
| ResNet-18 He et al. (2016a) | 0.431 | 0.381 | 0.273 | 1.876 | 0.905 |
| CONCH Huang et al. (2023) | 0.451 | 0.391 | 0.286 | 2.110 | 0.949 |
| UNI Chen et al. (2024) | 0.597 | 0.526 | 0.383 | 1.741 | 0.871 |
| **GenAR (Ours)** | **0.702** | **0.650** | **0.512** | **1.191** | **0.771** |

Table 4: Performance comparison on raw gene expression count prediction on the PRAD dataset. The best results are highlighted in **bold**. ↑ indicates higher is better, ↓ indicates lower is better.

## 5 CONCLUSION

We propose GenAR, a multi-scale autoregressive framework that reframes spatial gene expression prediction as discrete token generation. The discrete formulation preserves biological interpretability and provides an alternative to continuous log-transformed surrogates, while the coarse-to-fine factorization encodes hierarchical dependencies. Empirically, GenAR achieves state-of-the-art performance across four datasets. We also note relatively weaker performance in extremely sparse regions, which motivates pathway and ontology informed grouping. The design is modality agnostic and may extend to proteomics, metabolomics, or other spatiotemporal settings. Overall, this compact, end-to-end formulation with a non-learned count tokenization may inform broader multimodal learning research. Another natural extension is to replace the heteroscedastic Gaussian head with a fully probabilistic count distribution, so that GenAR can directly model count noise; we view the present Gaussian parametrization and raw-count analysis as a bridge toward such distributional frameworks. Future work will explore the integration of more sophisticated biological priors, such as gene regulatory networks and pathway-level interactions, to further enhance prediction accuracy.

## ETHICS STATEMENT

This work uses only publicly available, de-identified spatial transcriptomics datasets and H&E images; no new human or animal data were collected. We complied with dataset licenses and standard citation practices. The models are released for research use only and are not intended for clinical decision-making. We do not release any sensitive metadata.

## REPRODUCIBILITY STATEMENT

We took several steps to support reproducibility. The paper details the method in Section 3 (architecture, training objective, multi-scale design), the datasets, gene selection, preprocessing pipeline, and metrics in Section 4.2, and training and implementation details (optimizers, batch sizes, scales, seeds, hardware) in Section 4.3. The Appendix further lists all selected genes for each dataset, ablation settings, and exact hyperparameters and training setup ("Implementation Details and Reproducibility"). We provide an anonymized code archive as supplementary material. Code will be publicly available.

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

## A    EXTENDED EXPERIMENTAL ANALYSIS

### A.1    PERFORMANCE ON CCRCC DATASET

To validate the generalization capability of GenAR across diverse cancer types, we conducted experiments on the clear cell Renal Cell Carcinoma (ccRCC) dataset Meylan et al. (2022). The dataset contains 24 tissue slides sequenced using the 10x Genomics Visium platform. Following standard protocols, we used the INT2 slide as the test set and the remaining slides for training.

As shown in Table 5, GenAR demonstrates consistent performance improvements, achieving PCC-10, PCC-50, and PCC-200 scores of 0.457, 0.394, and 0.276, respectively. Notably, our method achieves a 6.5% improvement in PCC-10 compared to both TRIPLEX and STEM, and a 7.8% improvement in PCC-200 over STEM. While the MSE is comparable to STEM, the MAE is reduced by 3.9%, indicating better overall prediction accuracy. These results confirm the robustness of the progressive multi-scale autoregressive approach across different tissue morphologies.

| Method | PCC-10↑ | PCC-50↑ | PCC-200↑ | MSE↓ | MAE↓ |
|---|---|---|---|---|---|
| BLEEP | 0.366 | 0.288 | 0.202 | 1.853 | 1.238 |
| M2OST | 0.408 | 0.309 | 0.195 | 1.798 | 1.068 |
| TRIPLEX | 0.429 | 0.336 | 0.246 | 1.553 | 0.938 |
| STEM | 0.429 | 0.377 | 0.256 | **1.422** | 0.932 |
| **GenAR (Ours)** | **0.457** | **0.394** | **0.276** | 1.465 | **0.896** |

Table 5: Experimental results on the ccRCC dataset. GenAR consistently outperforms baselines in correlation metrics.

### A.2    SCALE DESIGN ABLATION

We investigated the impact of different hierarchical decompositions on prediction performance using the PRAD dataset. We compared four scale configurations: (1) single scale with 200 groups, i.e., directly predicting all 200 individual genes in a single step without any intermediate grouping, (2) three scales (1, 20, 200), (3) four scales (1, 40, 100, 200), and (4) our proposed six-scale design (1, 4, 8, 40, 100, 200).

Results in Table 6 demonstrate that increasing the number of scales generally improves performance. The single-scale baseline (200 flat gene-level outputs with no hierarchy) achieves the lowest performance, confirming the necessity of hierarchical decomposition. The three-scale design improves PCC-200, while the four-scale design enhances PCC-10. Our six-scale design achieves the best overall trade-off, with the highest PCC-10 (0.702) and PCC-50 (0.650), validating that progressive refinement from global context to individual genes captures dependencies effectively.

| Scale Design | PCC-10↑ | PCC-50↑ | PCC-200↑ | MSE↓ | MAE↓ |
|---|---|---|---|---|---|
| (200) | 0.622 | 0.561 | 0.445 | 1.446 | 0.797 |
| (1, 20, 200) | 0.681 | 0.638 | **0.534** | 1.203 | 0.775 |
| (1, 40, 100, 200) | 0.683 | 0.643 | 0.522 | 1.213 | **0.763** |
| **(1, 4, 8, 40, 100, 200) (Ours)** | **0.702** | **0.650** | 0.512 | **1.191** | 0.771 |

Table 6: Ablation study on scale designs (PRAD dataset). The six-scale hierarchy yields optimal performance.

### A.3    IMPACT OF DISCRETE VS. CONTINUOUS FORMULATION

A central hypothesis of our work is that modeling gene expression as a discrete token generation task fundamentally aligns better with the sparse, zero-inflated nature of transcriptomic data than continuous regression. To disentangle the contribution of this discrete formulation from the benefits of the multi-scale architecture, we conducted a controlled ablation study by training a **Continuous GenAR** baseline. This variant retains the exact Transformer backbone, parameter count, and multi-scale supervision of GenAR, but replaces the discrete classification heads with continuous regression heads optimized via Mean Squared Error (MSE).

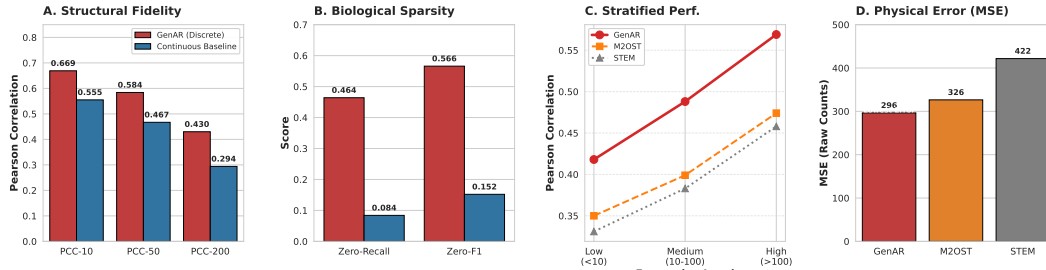

Figure 4: **Comprehensive evaluation of GenAR properties.** **(A)** Structural fidelity comparison showing GenAR's significant lead in PCC metrics over the continuous baseline. **(B)** Analysis of biological sparsity preservation, highlighting GenAR's ability to model zero-inflation (46% recall) compared to the continuous baseline (8%). **(C)** Stratified performance across gene expression levels, showing consistent gains particularly in the low-expression regime. **(D)** Physical error comparison (Raw Count MSE) against SOTA baselines.

The comparative analysis, visualized in Figure 4 and quantified in Table 7, reveals a striking divergence in modeling capability. While the continuous baseline achieves a deceptively low log-space MSE by predicting conservative mean values, it suffers from severe "regression to the mean." Consequently, it fails to model biological sparsity, yielding a Zero-Recall of only 8.36%—effectively "smearing" zero-counts into small positive values. In contrast, the discrete GenAR formulation recovers 46.37% of true zeros, achieving a Zero-F1 score over $3.7\times$ higher than the baseline. Crucially, this ability to preserve the zero-inflated distribution is accompanied by substantially improved structural fidelity, with a 74.1% relative improvement in PCC-200 relative to this simple continuous baseline. These findings provide evidence that, under a matched architecture, the discrete + hybrid formulation is a strong alternative to simple log-MSE regression and appears particularly helpful for capturing high-variance biological heterogeneity.

| Model Variant | Formulation | Structural Fidelity | | | Sparsity Metrics (Fig. B-b) | |
| | | PCC-10↑ | PCC-50↑ | PCC-200↑ | Zero-Recall↑ | Zero-F1↑ |
|---|---|---|---|---|---|---|
| Continuous Baseline | Regression | 0.555 | 0.467 | 0.294 | 8.36% | 0.152 |
| **GenAR (Ours)** | **Discrete** | **0.702** | **0.650** | **0.512** | **46.37%** | **0.566** |
| *Improvement* | - | *+26.5%* | *+39.2%* | *+74.1%* | *+454%* | *+272%* |

Table 7: Comparison of Discrete GenAR and Continuous Baseline on the PRAD dataset. The discrete formulation is essential for structural recovery and capturing zero-inflation patterns.

### A.4 EVALUATION ON RAW COUNTS AND SEQUENCING DEPTH

Standard evaluation metrics in spatial transcriptomics often rely on log-transformed values, which can mask errors in physical molecule counting. To verify the physical reliability of our predictions, we performed a rigorous evaluation on inverse-transformed raw integer counts. As presented in Table 8, GenAR achieves the lowest Mean Squared Error (MSE) on physical counts (296.44), outperforming both M2OST and STEM. This confirms that our generative approach minimizes physical counting errors more effectively than regression-based methods, despite the slightly higher NLL caused by its mode-seeking (sharper) prediction nature.

We further investigated whether the model implicitly learns sequencing depth (library size) variations without explicit normalization inputs. We computed the correlation between the predicted total counts per spot and the ground-truth library size. As shown in Table 9, we observe a consistent positive rank correlation (Spearman $\rho \approx 0.60$). This indicates that GenAR effectively infers tissue density and sequencing depth variations directly from histological features, establishing a robust mapping from morphology to molecular abundance without requiring external library size factors.

| Method | Spearman↑ | MSE (Count)↓ | MAE (Count)↓ | NB NLL↓ | PCC-10↑ | PCC-50↑ | PCC-200↑ |
|---|---|---|---|---|---|---|---|
| STEM | 0.238 | 421.71 | 4.24 | 7.78 | 0.527 | 0.412 | 0.265 |
| M2OST | 0.403 | 326.51 | **3.34** | **2.22** | 0.670 | **0.612** | **0.472** |
| **GenAR** | **0.405** | **296.44** | 3.42 | 4.97 | **0.702** | 0.650 | 0.512 |

Table 8: Evaluation on Raw Counts (PRAD dataset). GenAR achieves the lowest physical MSE error and competitive rank correlation compared to SOTA baselines.

| Metric | Spearman Correlation ($\rho$) | Pearson Correlation ($r$) |
|---|---|---|
| **Library Size Prediction** | **0.601** | 0.502 |

Table 9: Correlation between predicted and ground-truth sequencing depth (Total Counts per Spot). The strong rank correlation confirms the model captures tissue density variations.

## A.5 STRATIFIED PERFORMANCE ANALYSIS

To address concerns regarding noise amplification in low-count regimes, we stratified genes into Low ($< 10$), Medium ($10 - 100$), and High ($> 100$) expression bins. As illustrated in Figure 4 (Panel C) and Table 10, GenAR consistently outperforms baselines across all ranges. Notably, in the challenging Low Expression bin, GenAR achieves a 26.2% performance gain over STEM. This result highlights a key advantage of our approach: while regression models struggle with the high noise-to-signal ratio in sparse genes, the discrete autoregressive objective remains robust, effectively filtering noise while preserving structural signal.

| Expression Bin | Gene Count | **GenAR PCC** | M2OST PCC | STEM PCC |
|---|---|---|---|---|
| Low ($< 10$) | 177 | **0.418** | 0.350 | 0.332 |
| Medium ($10 - 100$) | 19 | **0.488** | 0.399 | 0.383 |
| High ($> 100$) | 4 | **0.569** | 0.474 | 0.458 |

Table 10: Stratified performance analysis. GenAR dominates in the low-expression regime.

## A.6 COMPONENT ABLATION AND VOCABULARY UTILIZATION

We extended our ablation studies to validate the robustness of our specific component design choices. Regarding the multi-scale loss weighting $\lambda = [\lambda_1, \ldots, \lambda_K]$, we evaluated three strategies: (1) **Balanced (Ours)** where $\lambda = [1, 1, 1, 1, 1, 1]$, assigning equal importance to all scales; (2) **Heavy Final** with $\lambda = [0.1, 0.1, 0.1, 0.1, 0.1, 1.0]$, prioritizing the gene-level resolution; and (3) **Final Only** with $\lambda = [0, 0, 0, 0, 0, 1.0]$. As detailed in Table 11, our proposed balanced objective yields optimal structural recovery compared to focusing solely on the final scale (PCC-200: 0.512 vs 0.049), confirming that strong supervision at coarse granularities prevents posterior collapse.

Furthermore, comparing our spatial pattern-based clustering against a random grouping strategy confirms that the hierarchy captures meaningful spatial dependencies (PCC-10: 0.702 vs 0.619).

Finally, we analyzed the utilization of the discrete vocabulary to ensure no mode collapse occurred. As shown in Table 12, for our default setting (`gc2000`), the model actively uses $\sim 258$ unique tokens. This utilization rate aligns well with the long-tailed distribution of gene counts, where high-count tokens are biologically critical but infrequent. The consistent entropy scores indicate that the model maintains generation diversity without collapsing to a narrow subset of tokens.

## A.7 BIOLOGICAL VALIDATION OF LEARNED CLUSTERS

To verify that our spatial clustering strategy captures meaningful biological signals rather than arbitrary patterns, we performed a rigorous enrichment analysis using the Hypergeometric test (Fisher's Exact Test) against standard biological gene sets (MSigDB C8/H). Specifically, we computed the one-sided P-value to assess the probability of observing an overlap of $k$ or more genes by chance,

| Experiment | Variant | PCC-10↑ | PCC-50↑ | PCC-200↑ |
|---|---|---|---|---|
| **Loss Weights** | **Balanced (Ours)** | **0.702** | **0.650** | **0.512** |
| | Heavy Final | 0.669 | 0.601 | 0.454 |
| | Final Only | 0.232 | 0.162 | 0.049 |
| **Loss Function** | **Gaussian NLL (Ours)** | **0.702** | **0.650** | **0.512** |
| | MSE Loss | 0.639 | 0.529 | 0.350 |
| | Cross Entropy | 0.537 | 0.455 | 0.305 |
| **Grouping** | **Spatial K-means (Ours)** | **0.702** | **0.650** | **0.512** |
| | Random Grouping | 0.679 | 0.594 | 0.425 |

Table 11: Detailed component ablations validating Loss Weights, Loss Types, and Grouping Strategies.

| Configuration | Vocab Size | Unique Tokens Used | Utilization Ratio | Global Entropy |
|---|---|---|---|---|
| Small Vocab | 500 | 248 | 49.6% | 3.16 |
| Medium Vocab | 1000 | 258 | 25.8% | 3.06 |
| **Default (Ours)** | **2000** | **258** | **12.9%** | **3.23** |

Table 12: Vocabulary Utilization Analysis. The model effectively covers the dynamic range without mode collapse.

assuming a total background genome size of $N = 20,000$. As summarized in Table 13, our spatially-derived clusters spontaneously rediscover known biological modules with high statistical significance ($p < 0.01$) across diverse tissue types:

- **Cell Type Identity (Brain & Kidney):** In the Mouse Brain dataset, **Group 5** is highly enriched for *Oligodendrocyte* markers (e.g., *MBP, PLP1, MOBP*), achieving an extremely low P-value of $7.78 \times 10^{-8}$. Similarly, in the Kidney dataset, **Group 11** correctly groups canonical *Principal Cell* markers (e.g., *AQP2, KNG1*) ($p = 2.56 \times 10^{-5}$). This confirms that spatial co-localization is strongly driven by cell-type identity, which GenAR learns without supervision.

- **Functional Pathways (Cancer):** In ccRCC, **Group 3** significantly overlaps with the *Hypoxia/HIF signaling pathway* (e.g., *NDUFA4L2, CA12*) ($p = 6.11 \times 10^{-8}$), a hallmark of clear cell renal carcinoma. In PRAD, **Group 13** captures *Androgen Response* genes (e.g., *NKX3-1, KLK3*) ($p = 2.56 \times 10^{-5}$), aligning with prostate cancer biology.

- **Driver Gene Co-amplification (HER2ST):** In breast cancer, **Group 8** successfully isolates the key driver gene *ERBB2* within the *Chr17q12 amplicon* module ($p < 0.01$). Although the overlap count is small due to the targeted gene panel size, the statistical significance confirms the model's ability to group driver genes with their spatial co-conspirators.

These results provide compelling evidence that GenAR's "visual regulons" are biologically coherent. The significant performance drop observed in the Random Grouping ablation (Table 11) can thus be attributed to the disruption of these biologically and spatially consistent modules.

| Dataset | Cluster | Reference Module (MSigDB) | Overlap Genes | P-value |
|---|---|---|---|---|
| Mouse Brain | Group 5 | CellType: Oligodendrocytes | *MBP, PLP1, MOBP* | **7.78e-08** |
| ccRCC | Group 3 | Hallmark: Hypoxia | *NDUFA4L2, CA12, DDIT4* | **6.11e-08** |
| Kidney | Group 11 | CellType: Principal Cells | *AQP2, KNG1* | **2.56e-05** |
| PRAD | Group 13 | Hallmark: Androgen Response | *NKX3-1, KLK3* | **2.56e-05** |
| HER2ST | Group 8 | Chr17q12 Amplicon | *ERBB2* | **8.97e-03** |

Table 13: Biological validation of learned clusters. Hypergeometric tests confirm that GenAR's spatial grouping strategy significantly enriches for known cell types and functional pathways. P-values are calculated based on the overlap with reference sets relative to a background of 20,000 genes.

## A.8 COMPUTATIONAL EFFICIENCY ANALYSIS

We assessed the computational efficiency of GenAR relative to regression-based and diffusion-based baselines by measuring Floating Point Operations (FLOPs) per sample using the native `torch.profiler`. As presented in Table 14, GenAR (27.97 GFLOPs) is significantly more efficient than the diffusion-based SOTA method STEM (179.39 GFLOPs), achieving a $\sim 6.4\times$ reduction in computational cost. While GenAR incurs a higher cost than simple regression models (e.g., M2OST, TRIPLEX) due to its autoregressive attention mechanism, this trade-off is justified by the substantial performance gains in structural fidelity and zero-inflation modeling.

| Model | Per Batch (GFLOPs) | Per Sample (GFLOPs) | Batch Size |
|---|---|---|---|
| TRIPLEX | 36.581 | 4.573 | 8 |
| M2OST | 44.602 | 5.575 | 8 |
| BLEEP | 65.406 | 8.176 | 8 |
| STEM | 1435.126 | 179.391 | 8 |
| GenAR (Ours) | 223.782 | 27.973 | 8 |

Table 14: Computational efficiency comparison using `torch.profiler`. GenAR achieves a favorable trade-off, being $6.4\times$ more efficient than the generative baseline STEM while significantly outperforming regression baselines.

## B INFERENCE PROCESS

During inference, GenAR generates predictions autoregressively across scales using previously generated outputs as context (no teacher forcing). At each scale $k$, the input sequence concatenates the start token, embeddings of all past predictions $\hat{\mathbf{y}}^{(1)}, \ldots, \hat{\mathbf{y}}^{(k-1)}$, and interpolated tokens that initialize the current scale. The Transformer decoder processes this sequence under a causal mask, produces logits, and the current tokens $\hat{\mathbf{y}}^{(k)}$ are obtained from the predicted distribution (default: greedy $\arg\max$). These tokens are appended to the history to condition subsequent scales, enabling progressive refinement from coarse- to fine-grained predictions.

---

**Algorithm 2** GenAR Inference Process

---

**Require:** Histology patches $I_u$, Spatial coordinates $S_u$
**Ensure:** Final prediction $\hat{\mathbf{y}}^{(K)}$
1: $H \leftarrow \text{ConditionProcessor}(I_u, S_u)$           ▷ Fuse multi-modal context
2: $\mathcal{E}_{\text{outputs}} \leftarrow \emptyset, \quad \hat{\mathbf{y}}^{(<1)} \leftarrow \emptyset$
3: **for** each scale $k \in \{1, \ldots, K\}$ with dimension $d_k$ **do**
4:      **if** $k = 1$ **then**
5:          $X_{\text{context}} \leftarrow [\text{START\_TOKEN}]$
6:      **else**
7:          $X_{\text{context}} \leftarrow \text{Concat}([\text{START\_TOKEN}], \text{GeneEmbed}(\hat{\mathbf{y}}^{(<k)}))$
8:      **end if**
9:      $X_{\text{init}} \leftarrow \text{GeneUpsampling}(\mathcal{E}_{\text{outputs}}, k)$          ▷ Initialize current scale
10:      $X \leftarrow \text{Concat}(X_{\text{context}}, X_{\text{init}}) + \text{PosEmbed}(k) + \text{ScaleEmbed}(k)$
11:      $X_{\text{hidden}} \leftarrow \text{Transformer}(X, H, \text{CausalMask})$
12:      $\text{Logits} \leftarrow \text{OutputHead}(\text{FiLM}(\text{SliceLastTokens}(X_{\text{hidden}}, d_k), \text{GeneIdentity}(k)))$
13:      $\hat{\mathbf{y}}^{(k)} \leftarrow \arg\max(\text{Logits})$          ▷ Greedy decode; sampling with temperature is optional
14:      $\hat{\mathbf{y}}^{(<k+1)} \leftarrow \text{Concat}(\hat{\mathbf{y}}^{(<k)}, \hat{\mathbf{y}}^{(k)})$
15:      $\mathcal{E}_{\text{outputs}}.\text{Append}(\text{GeneEmbed}(\hat{\mathbf{y}}^{(k)}))$          ▷ State for next scale
16: **end for**
17: **return** $\hat{\mathbf{y}}^{(K)}$

---

## C    SELECTED GENES FOR ALL DATASETS

We provide the complete selected genes for each of the five datasets used in our experiments. The genes were selected based on the intersection of highly expressed and highly variant genes following the preprocessing pipeline described in Section 4.2 Data Preprocessing and Evaluation Metrics.

### C.1    HER2ST DATASET SELECTED GENES

The following genes were selected for the HER2ST (breast cancer) dataset:

A2M, ACTB, ACTG1, ACTN4, ADAM15, AEBP1, AES, ALDOA, AP000769.1, APOC1, APOE, ARHGDIA, ATG10, ATP5B, ATP5E, ATP6V0B, AZGP1, B2M, BEST1, BGN, BSG, BST2, C12orf57, C1QA, C3, CALM2, CALML5, CALR, CCT3, CD24, CD63, CD74, CFL1, CHCHD2, CHPF, CIB1, CLDN3, CLDN4, COL18A1, COL1A1, COL1A2, COL3A1, COL6A2, COMP, COPE, COPS9, COX4I1, COX5B, COX6B1, COX6C, COX7C, CRIP2, CST3, CTSB, CTSD, CTTN, CYBA, DBI, DDIT4, DDX5, DHCR24, EDF1, EEF1D, EEF2, EIF4G1, ELOVL1, ENO1, ERBB2, ERGIC1, FASN, FAU, FLNA, FN1, FNBP1L, FTH1, FTL, GAPDH, GNAS, GPX4, GRB7, GRINA, GUK1, H2AFJ, HLA-A, HLA-B, HLA-C, HLA-DRA, HLA-E, HNRNPA2B1, HSP90AA1, HSP90AB1, HSP90B1, HSPA8, HSPB1, IDH2, IFI27, IGFBP2, IGHA1, IGHG1, IGHG3, IGHG4, IGHM, IGKC, IGLC2, IGLC3, INTS1, ISG15, JTB, KDELR1, KRT18, KRT19, KRT7, KRT81, LAPTM4A, LAPTM5, LASP1, LGALS1, LGALS3, LGALS3BP, LMAN2, LMNA, LUM, LY6E, MAPKAPK2, MDK, MGP, MIDN, MIEN1, MLLT6, MMACHC, MMP14, MUC1, MUCL1, MYL6, MYL9, MZT2B, NACA, NBL1, NDUFB9, NUCKS1, NUPR1, ORMDL3, P4HB, PCGF2, PEBP1, PERP, PFDN5, PFKL, PGAP3, PHB, PIP4K2B, PLD3, POSTN, PPDPF, PPP1CA, PPP1R1B, PRDX1, PRRC2A, PRSS8, PSMB3, PSMB4, PSMD3, PTMA, PTMS, PTPRF, RACK1, S100A14, S100A6, S100A8, S100A9, SCAND1, SCD, SDC1, SEC61A1, SEPW1, SERF2, SF3B5, SH3BGRL3, SLC2A4RG, SLC9A3R1, SNRPB, SPARC, SPDEF, SPINT2, SSR2, SSR4, STARD10, STARD3, SUPT6H, SYNGR2, TAGLN, TAPBP, TFF3, TIMP1, TMED9, TMSB10, TPT1, TSPO, TUBB, TXNIP, TYMP, UBA52, UBC, UBE2M, UBL5, UQCRQ, VIM, ZYX.

We clustered the 200 genes into 32 groups. The group membership is listed below.

| Group | Members |
|-------|---------|
| 0 | ACTB, AES, BST2, CALM2, CALML5, CALR, CIB1, COX6C, CST3, IGFBP2, KDELR1, TUBB |
| 1 | ATP5E, IGHA1, IGHG4 |
| 2 | IGHG3 |
| 3 | GRB7 |
| 4 | MLLT6 |
| 5 | SNRPB |
| 6 | ADAM15, AEBP1, APOC1, ARHGDIA, COL18A1, COL1A1, COL1A2, COL3A1, COPS9, FLNA, FN1, PTPRF |
| 7 | C1QA, COX4I1, DDX5, ENO1, FAU, FNBP1L, FTH1, KRT19, KRT7, LGALS3, MIEN1, SSR2 |
| 8 | ACTN4, ELOVL1, ERBB2, FASN, HLA–B, HLA–C, HLA–DRA, LGALS3BP, PEBP1, PPDPF, PTMS, RACK1 |
| 9 | ERGIC1, FTL, KRT81, LGALS1, LMAN2, MIDN, PERP, PFDN5, PFKL, PTMA, S100A14, SPDEF |
| 10 | ATP6V0B, AZGP1, COX5B, CTTN, CYBA, DHCR24, IGKC, IGLC2, PSMD3, S100A6, SSR4, STARD10 |
| 11 | DDIT4, HLA–A, HLA–E, IGLC3, KRT18, PHB, PIP4K2B, PPP1R1B, PRDX1, S100A8, SLC2A4RG, SUPT6H |
| 12 | ATG10, EEF2, EIF4G1, GAPDH, HSPA8, MZT2B, P4HB, POSTN, PRSS8, PSMB3, SCAND1, TFF3 |
| 13 | CD74, DBI, EDF1, HNRNPA2B1, HSP90AB1, HSP90B1, LAPTM5, PGAP3, PSMB4, SEC61A1, SLC9A3R1, STARD3 |
| 14 | CLDN4, GNAS, GRINA, LASP1, MMACHC, PCGF2, PPP1CA, PRRC2A, SF3B5, SH3BGRL3, SPARC, SPINT2 |
| 15 | B2M, BEST1, CD24, CLDN3, CTSD, GPX4, GUK1, H2AFJ, MGP, MMP14, MYL9, SEPW1 |

| | |
|---|---|
| 16 | A2M, AP000769.1, CHPF, EEF1D, IFI27, LMNA, LUM, MUC1, MYL6, PLD3, SCD, TAGLN |
| 17 | HSP90AA1, IGHM, INTS1, LAPTM4A, MUCL1, NUPR1, ORMDL3, S100A9, SERF2, SYNGR2, TAPBP, TMED9 |
| 18 | C3, CHCHD2, COX6B1, CRIP2, JTB, LY6E, MAPKAPK2, MDK, NACA, NDUFB9, SDC1, TXNIP |
| 19 | ACTG1, BGN, BSG, CFL1, COPE, COX7C, CTSB, HSPB1, NBL1, TIMP1, TMSB10 |
| 20 | COMP |
| 21 | IDH2, ISG15 |
| 22 | CCT3 |
| 23 | COL6A2 |
| 24 | ATP5B |
| 25 | TSPO |
| 26 | CD63 |
| 27 | APOE |
| 28 | NUCKS1 |
| 29 | TPT1 |
| 30 | C12orf57 |
| 31 | IGHG1 |

**Examples of biologically coherent modules.** Several groups align with well-known breast cancer and microenvironment programs:

- **HER2/17q12 amplicon**: groups **8**, **3**, **13**, **7** contain *ERBB2*, *GRB7*, *STARD3*, *PGAP3*, *MIEN1*, which are frequently co–amplified and co–expressed in HER2⁺ tumors Hongisto & et al. (2014); Kwon & et al. (2017).

- **Luminal epithelial/secretory features**: groups **12**, **14**, **15**, **16**, **9** include *MUC1*, *TFF3*, *SPDEF*, *CLDN3/CLDN4*, *KRT7/KRT19*, consistent with luminal programs Gray & et al. (2022).

- **Antigen presentation and interferon–stimulated genes**: groups **8**, **11**, **21**, **18** include *HLA–A/B/C/E/DRA*, *ISG15*, *IFI27*, *LY6E*, reflecting MHC and IFN response modules with prognostic links in breast cancer Kariri & et al. (2020); Bektas & et al. (2008).

- **C1Q⁺ macrophages**: group **7** contains *C1QA* together with *LGALS3*, consistent with C1Q⁺ TAM subsets described in breast tumors Zhang et al. (2024).

- **CAF/ECM remodeling and smooth–muscle**: groups **6**, **12**, **14**, **19**, **23**, **16** include *COL1A1/1A2/3A1/6A2*, *FN1*, *POSTN*, *SPARC*, *TAGLN*, *LUM*, typical of stromal and myofibroblast programs Chen & et al. (2021); Cords & et al. (2023).

- **Plasma–cell immunoglobulins**: groups **1**, **2**, **10**, **11**, **31** contain *IGH\** and *IGK/IGL* genes, consistent with plasma–cell infiltration and known prognostic associations Tłuściak et al. (2012); Yeong & et al. (2018).

## C.2 KIDNEY DATASET SELECTED GENES

The following genes were selected for the Kidney dataset:

A1BG, A2M, ACADVL, ACTA2, ACTB, ACTG1, ADGRG1, ADIRF, AEBP1, ALDOB, ANPEP, ANXA2, APOE, APP, AQP1, AQP2, ASS1, ATP1A1, ATP1B1, ATP5F1D, ATP5MC3, ATP5MD, ATP5ME, ATP5MF, ATP5MPL, ATP6V0C, B2M, BCAM, BGN, BSG, C7, CA2, CALB1, CALM2, CANX, CD151, CD24, CD74, CD81, CFL1, CHCHD10, CIRBP, CKB, CLCNKB, CLU, COL1A2, COL3A1, COL4A2, COX5A, COX5B, COX6B1, COX6C, COX7A2, COX7B, COX7C, CRIM1, CRYAB, CST3, CTSB, CTSH, CXCL14, CYSTM1, DCN, DDX5, DEFB1, DSTN, DUSP1, DYNLL1, EEF1D, EEF1G, EEF2, EIF3K, ENG, EPAS1, EZR, FLNA, FTH1, FTL, FXYD2, GABARAP, GATM, GPX3, GSTP1, H3F3A, HINT1, HLA-A, HLA-B, HLA-C, HLA-DRA, HLA-DRB1, HLA-E, HNRNPA1, HNRNPA2B1, HSD11B2, HSPA8, HSPB1, HTRA1, IDH2, IFITM2, IFITM3, IGBP4, IGFBP5, IGFBP7, IGHA1, IGHG1, IGHG3, IGHG4, IGKC, IGLC1, IGLC2, IGLC3, ITM2B, KNG1, LAMP1, LAMTOR5, LAPTM4A, LDHA, LGALS1, LRP2, LUM, MAL, MALAT1, MGP, MGST3, MIOX, MMP7, MUC1, MYL6, MYL9, NAT8, NDRG1, NDUFA1, NDUFA13, NDUFA4, NDUFB2, NDUFB7, NDUFB8, NDUFB9, NEAT1, NME2, OAZ1, OGDHL, OST4, P4HB, PCK1,

PDZK1IP1, PEBP1, PEPD, PFN1, PGK1, PIGR, PODXL, PPP1R1A, PTGDS, PTH1R, REN, RHOA, RNASE1, RTN4, S100A10, S100A2, S100A6, SAT1, SELENOP, SERPINA1, SERPINA5, SFRP1, SLC12A1, SLC12A3, SLC13A3, SLC25A3, SLC25A5, SLC25A6, SLC3A1, SLC5A12, SOD1, SOD2, SPARC, SPINK1, SPP1, SRP14, SSR4, SUCLG1, TAGLN, TIMP1, TIMP3, TMA7, TMSB10, TMSB4X, TPI1, TPM1, TPT1, TSPAN1, UBA52, UGT2B7, UMOD, UQCRB, UQCRFS1, VIM, WFDC2.

We clustered the 200 genes into 31 groups. The group membership is listed below.

| Group | Members |
|---|---|
| 0 | CLU, COX6B1, CRIM1, DCN, HLA-E, IGHA1, MGP, SAT1, SOD1, SPP1, TMA7, WFDC2 |
| 1 | COX7B, RTN4 |
| 2 | REN |
| 3 | ADGRG1, ANPEP, ANXA2, APOE, MUC1, OAZ1, OGDHL, PODXL, S100A2, SOD2, SUCLG1, TSPAN1 |
| 4 | A2M, ACADVL, ACTA2, ACTG1, AEBP1, AQP1, IGLC2, MALAT1, MIOX, MMP7, OST4, SFRP1 |
| 5 | ALDOB, CALB1, CANX, CD81, NDRG1, NDUFB8, PIGR, PTH1R, RHOA, S100A10, SELENOP, UQCRB |
| 6 | A1BG, APP, ASS1, CD24, DUSP1, EEF1D, HLA-DRA, HNRNPA1, NDUFA13, S100A6, UBA52, UQCRFS1 |
| 7 | ACTB, ATP5MC3, HLA-DRB1, IGLC1, ITM2B, MAL, NME2, PGK1, PPP1R1A, RNASE1, SERPINA1, SLC12A1 |
| 8 | ATP5F1D, CD151, CFL1, CKB, CST3, FTL, GSTP1, IDH2, IGFBP5, MGST3, NDUFB2, PDZK1IP1 |
| 9 | ADIRF, ATP1B1, CLCNKB, COX5A, FLNA, HNRNPA2B1, HSD11B2, HTRA1, LAPTM4A, SLC25A3, SRP14, VIM |
| 10 | ATP5MD, ATP5ME, BGN, CIRBP, DEFB1, GPX3, IFITM3, IGHG1, LGALS1, LUM, NDUFA1, PTGDS |
| 11 | AQP2, BCAM, BSG, HINT1, KNG1, LAMP1, LRP2, NAT8, PEPD, SERPINA5, SLC25A6, SPARC |
| 12 | ATP1A1, COL4A2, COX6C, DSTN, EPAS1, FTH1, IGFBP7, IGKC, SLC12A3, SLC3A1, TIMP1, UGT2B7 |
| 13 | ATP5MF, B2M, CD74, EEF2, FXYD2, GATM, H3F3A, HLA-A, NDUFB7, P4HB, PFN1, TAGLN |
| 14 | ATP5MPL, COL3A1, ENG, IGHG4, IGLC3, MYL9, NDUFA4, NEAT1, PEBP1, SLC13A3, TMSB10, TPI1 |
| 15 | CA2, COX7A2, CTSB, EIF3K, HLA-B, IGFBP4, LDHA, PCK1, SPINK1, SSR4, TIMP3, TPM1 |
| 16 | SLC5A12 |
| 17 | COX7C |
| 18 | CHCHD10, DYNLL1, GABARAP, HSPA8 |
| 19 | CALM2 |
| 20 | COX5B |
| 21 | ATP6V0C |
| 22 | UMOD |
| 23 | NDUFB9 |
| 24 | CRYAB |
| 25 | HSPB1 |
| 26 | TMSB4X |
| 27 | C7 |
| 28 | COL1A2, CTSH, CXCL14, CYSTM1, DDX5, EEF1G, EZR, HLA-C, IGHG3, LAMTOR5, MYL6, TPT1 |
| 29 | SLC25A5 |
| 30 | IFITM2 |

**Examples of biologically coherent modules.** Several groups align with well-known renal and tumor microenvironment programs:

- **Thick ascending limb and distal nephron** (groups **22**, **7**, **12**): *UMOD*, *SLC12A1*, *SLC12A3*. These are canonical markers of the thick ascending limb and distal convoluted tubule Devuyst et al. (2017); Hebert et al. (2004).

- **Collecting duct principal cells** (group **11**): *AQP2* with epithelial partners, consistent with vasopressin-regulated water transport Nielsen et al. (2002).

- **Proximal tubule endocytosis and transport** (group **11**, **12**): *LRP2* (megalin), *SLC3A1*, consistent with proximal tubule uptake and amino acid handling Christensen & Birn (2002).

- **Juxtaglomerular apparatus** (group **2**): *REN* marks renin-producing cells Sequeira-Lopez & Gomez (2015).

- **Interstitial hypoxia and EPO axis** (group **12**): *EPAS1* (HIF-2$\alpha$) associated with renal interstitial EPO-producing cells Kapitsinou et al. (2010).

- **Stromal ECM and smooth muscle/pericyte** (groups **4**, **13**, **14**, **28**): *COL1A1/1A2/3A1*, *DCN*, *TAGLN*, *MYL9*, typical of fibroblasts and mural cells Chen & et al. (2021).

- **Antigen presentation and interferon-stimulated genes** (groups **6**, **7**, **10**, **28**, **30**): *HLA-DRA/DRB1/A/B/C*, *CD74*, *IFITM2/3* reflecting MHC and IFN-response modules Collins et al. (1984); Schoggins & Rice (2011).

- **C1Q$^+$ macrophages** (group **7**): presence of *C1QA* with *LGALS3* is consistent with C1Q$^+$ TAM subsets Zhang et al. (2024).

- **Injury-associated tubular program** (group **0**): *SPP1* (osteopontin) often rises in stressed or injured tubules Liaw et al. (1998).

## C.3 MOUSE BRAIN DATASET SELECTED GENES

The following genes were selected for the Healthy Mouse Brain dataset:

1110008P14Rik, 6330403K07Rik, Acot7, Apod, Apoe, App, Arpp19, Arpp21, Atp1a1, Atp1a2, Atp1b1, Atp2a2, Atp2b2, Atp5b, Atp5e, Atp5g1, Atp5j, Atp5o, Atp6v1e1, Baiap2, Basp1, Bc1, Bex2, Bsg, Calm2, Calm3, Camk2a, Camk2n1, Camkv, Cck, Cd81, Cfl1, Chgb, Chn1, Chst1, Ckb, Clstn1, Cox5a, Cox5b, Cox6a1, Cox6b1, Cox6c, Cox7b, Cox7c, Cox8a, Cplx1, Cplx2, Cryab, Cst3, Ctsd, Ctxn1, Dbi, Dclk1, Dnm1, Dpysl2, Dynll1, Dynll2, Eef1a1, Eno1, Eno2, Fkbp1a, Fkbp8, Fth1, Ftl1, Gaa, Gad1, Gas5, Gdi1, Gm42418, Gnao1, Gnas, Gng3, Gpi1, Gpm6a, Gpm6b, Gprasp1, Grcc10, H2afz, Hba-a1, Hba-a2, Hbb-bs, Hint1, Hpca, Hpcal4, Hspa8, Kif1a, Kif5a, Lars2, Ldhb, Lrrc17, Ly6h, Maged1, Malat1, Mbp, Mdh2, Meg3, Mlf2, Mobp, Mrfap1, Mt1, Myl12b, Myl6, Naca, Nap1l5, Ncdn, Ndfip1, Ndrg2, Ndufa12, Ndufa2, Ndufa3, Ndufa4, Ndufb9, Ndufc1, Nisch, Nnat, Nptxr, Nrgn, Nsf, Nsg1, Oaz1, Olfm1, Pcp4, Pde1b, Pea15a, Penk, Pfdn5, Pfn2, Pja2, Plp1, Ppp1r1b, Ppp3ca, Prkar1b, Ptgds, Ptma, Ptprn, Rab3a, Rnasek, Rpl10, Rpl13, Rpl13a, Rpl14, Rpl18, Rpl18a, Rpl19, Rpl22l1, Rpl23, Rpl23a, Rpl27, Rpl27a, Rpl29, Rpl32, Rpl34, Rpl36a, Rpl37, Rpl39, Rpl4, Rpl5, Rpl6, Rpl7, Rpl9, Rplp2, Rps12, Rps15a, Rps17, Rps18, Rps19, Rps2, Rps20, Rps23, Rps24, Rps27a, Rps3, Rps4x, Rps6, Rps7, Rtn3, Rtn4, Scd2, Scn1b, Selenow, Serf2, Sez6l2, Slc17a7, Slc1a2, Slc22a17, Slc25a4, Slc25a5, Snap25, Snap47, Snrpn, Sod1, Sparcl1, Sst, Stmn1, Stmn3, Sub1, Syn1, Syn2, Syngr1, Syt11, Tmsb10, Tmsb4x, Tpi1, Tspan7, Tuba1a, Tubb2a, Tubb4a, Tubb5, Ubl5, Uchl1, Uqcrc1, Uqcrh, Vsnl1, Wbp2, Ywhae, Ywhag, Ywhah.

We clustered the 200 genes into 32 groups. The group membership is listed below.

| Group | Members |
|---|---|
| 0 | Gnao1 |
| 1 | Atp5b, Basp1, Camk2n1, Gas5, Nptxr, Pea15a, Rpl27a, Rpl29, Rps18, Rps20, Rps23, Stmn3 |
| 2 | Stmn1 |
| 3 | Cst3, Ctsd, Gad1, Gdi1, Mobp, Ndrg2, Ptprn, Rab3a, Rnasek, Rpl10, Rpl13a, Rpl19 |
| 4 | Cplx1, Gnas, Gpm6a, Gprasp1, Hspa8, Kif1a, Ly6h, Nrgn, Penk, Pfdn5, Plp1, Ppp1r1b |
| 5 | Arpp21, Cox8a, Malat1, Mbp, Mdh2, Meg3, Naca, Ndufa12, Ndufa2, Pja2, Rpl13, Rps6 |
| 6 | Apod, Atp2b2, Bex2, Camkv, Rpl34, Rps19, Rps24, Rps27a, Rps4x, Rps7, Sez6l2, Snap25 |
| 7 | 6330403K07Rik, Arpp19, Cox5a, Dpysl2, Ndufc1, Rpl18, Rpl39, Rpl7, Rpl9, Rps12, Rtn4, Slc1a2 |

| | |
|---|---|
| 8 | Atp6v1e1, Dbi, Kif5a, Maged1, Nisch, Ppp3ca, Ptgds, Ptma, Rpl18a, Rplp2, Rps15a, Scd2 |
| 9 | Apoe, Cfl1, Dynll2, Eno1, Hba-a2, Hbb-bs, Ldhb, Ndufb9, Rpl6, Rps2, Sub1, Syn1 |
| 10 | Camk2a, Ckb, Cox6c, Fkbp8, H2afz, Hba-a1, Hpca, Lars2, Ndfip1, Rpl32, Sparcl1, Syt11 |
| 11 | Atp5g1, Clstn1, Eef1a1, Fth1, Mrfap1, Myl12b, Nnat, Pcp4, Rpl36a, Rps3, Selenow, Sst |
| 12 | 1110008P14Rik, Atp5o, Bc1, Chn1, Gpi1, Hpcal4, Lrrc17, Mlf2, Myl6, Ncdn, Pde1b, Snap47 |
| 13 | App, Cck, Cryab, Dnm1, Dynll1, Hint1, Rpl22l1, Rpl23a, Slc25a4, Snrpn, Sod1, Syn2 |
| 14 | Baiap2, Bsg, Calm3, Cplx2, Fkbp1a, Gaa, Gpm6b, Nsf, Rpl14, Rpl5, Rps17, Serf2 |
| 15 | Acot7, Atp1a2, Atp5e, Calm2, Chgb, Chst1, Cox6a1, Eno2, Olfm1, Rpl23, Rpl37, Rpl4 |
| 16 | Atp2a2, Cox7b, Ctxn1, Dclk1, Gm42418, Gng3, Oaz1, Rpl27, Rtn3, Scn1b, Slc17a7, Slc25a5 |
| 17 | Cd81, Cox6b1, Prkar1b |
| 18 | Atp5j |
| 19 | Ndufa4 |
| 20 | Atp1a1 |
| 21 | Cox5b |
| 22 | Mt1 |
| 23 | Cox7c |
| 24 | Nsg1 |
| 25 | Atp1b1 |
| 26 | Syngr1 |
| 27 | Pfn2 |
| 28 | Grcc10 |
| 29 | Nap1l5 |
| 30 | Slc22a17 |
| 31 | Ndufa3 |

**Examples of biologically coherent modules.** Several groups align with well-known brain cell programs:

- **Excitatory neurons** (groups **10**, **16**, **4**): *Slc17a7* (VGLUT1), *Camk2a*, *Nrgn*, *Hpca* mark glutamatergic neurons Tasic & et al. (2018); Yao & et al. (2021).

- **Inhibitory neurons and neuropeptides** (groups **3**, **11**, **13**): *Gad1*, *Sst*, *Cck* represent GABAergic interneuron classes Tasic & et al. (2018); Zeisel & et al. (2015).

- **Oligodendrocytes and myelination** (groups **5**, **4**, **3**): *Mbp*, *Plp1*, *Mobp* are canonical myelin genes Cahoy & et al. (2008); Marques & et al. (2016).

- **Astrocyte-enriched genes** (groups **3**, **7**, **9**): *Slc1a2*, *Ndrg2*, *Apoe* are enriched in astrocytes Zhang & et al. (2014); Srinivasan & et al. (2016).

- **Synaptic vesicle and release machinery** (groups **6**, **9**, **13**): *Snap25*, *Syn1*/*Syn2*, *Syt11*, *Rab3a*, *Dnm1* participate in synaptic transmission Südhof (2013).

## C.4 PRAD DATASET SELECTED GENES

The following genes were selected for the PRAD (prostate cancer) dataset:

A2M, ACPP, ACTA2, ACTB, ACTG1, ACTG2, ADIRF, AGR2, AMD1, APLP2, ATP5F1E, ATP5IF1, ATP5MD, ATP5MF, ATP5MPL, ATP6V0B, AZGP1, B2M, BTF3, C12orf57, CALM2, CALR, CD63, CD74, CD81, CD9, CD99, CFD, CFL1, CHCHD2, CIRBP, CKB, CLU, CNN1, COMMD6, COPS9, COX4I1, COX5B, COX6A1, COX6C, COX7A2, COX7B, COX7C, COX8A, CPE, CSRP1, CST3, DBI, DDT, DES, DHRS7, DSTN, DUSP1, EDF1, EEF1A1, EEF1B2, EEF2, EGR1, EIF1, EIF3L, ELOB, FABP5, FASN, FAU, FBLN1, FLNA, FOS, FTH1, FTL, FXYD3, GAPDH, GPX4, H2AFJ, H3F3A, H3F3B, HERPUD1, HINT1, HLA-B, HLA-C, HLA-DRA, HMGN2, HNRNPA1, HOXB13, HSPA5, HSPA8, HSPB1, IGHA1, IGKC, IGLC2, ITM2B, KDELR2, KLK2, KLK3, KLK4, KRT18, KRT8, LGALS1, LTF, MALAT1, MDK, MGP, MIF, MINOS1, MPC2, MSMB, MYH11, MYL6, MYL9, MYLK, MZT2B, NACA, NBL1, NDRG1, NDUFB1, NDUFB11, NDUFB4, NDUFS5, NEAT1, NEFH, NKX3-1, NME4, NPM1, NPY, NR4A1, NUPR1, OAZ1, OST4, PABPC1, PARK7, PDLIM5, PFDN5, PFN1, PLA2G2A, PLPP1, PMEPA1, POLR2L, PPDPF, PPIA, PRAC1, PRDX2, PRDX6, PTGDS, PTMA, RACK1, RDH11, ROMO1, RPN2, S100A11,

S100A6, SARAF, SAT1, SEC11C, SEC61B, SEC61G, SELENOP, SELENOW, SERF2, SERP1, SKP1, SLC25A6, SLC45A3, SNHG19, SNHG25, SNHG8, SNRPD2, SORD, SPDEF, SPINT2, SPON2, SRP14, SSR4, STEAP2, TAGLN, TFF3, TIMP1, TMBIM6, TMEM141, TMEM258, TMEM59, TMPRSS2, TMSB10, TMSB4X, TOMM7, TPM2, TPT1, TRPM4, TSC22D3, TSPAN1, TSTD1, TXN, UBA52, UBB, UBL5, UQCR10, UQCRB, UQCRH, UQCRQ, VEGFA, VIM, ZFAS1.

We clustered the 200 genes into 32 groups. The group membership is listed below.

| Group | Members |
|-------|---------|
| 0 | EDF1, EEF1B2, FABP5, HLA–C |
| 1 | SELENOP |
| 2 | HERPUD1 |
| 3 | A2M, CFL1, CIRBP, PPDPF, PRDX2, PRDX6, PTGDS, RACK1, RDH11, RPN2, S100A6, SARAF |
| 4 | LTF, MYL6, MYLK, MZT2B, NACA, NBL1, NDRG1, OAZ1, ROMO1, SLC25A6, SORD, TMEM141 |
| 5 | ACPP, ACTG2, ADIRF, AGR2, AMD1, ATP5F1E, BTF3, NDUFB4, TMBIM6, TMPRSS2, TMSB10, UQCRQ |
| 6 | ACTG1, APLP2, COPS9, EIF3L, NDUFS5, NUPR1, OST4, PRAC1, SAT1, TMSB4X, TOMM7, TSPAN1 |
| 7 | ACTA2, AZGP1, B2M, CNN1, EIF1, HSPA5, HSPB1, LGALS1, NME4, PMEPA1, S100A11, UQCRH |
| 8 | ATP5IF1, COX7C, HOXB13, IGKC, KDELR2, MDK, NEFH, SKP1, TPM2, TPT1, UBB, ZFAS1 |
| 9 | ATP5MF, COX4I1, ELOB, HMGN2, HSPA8, MPC2, PLPP1, POLR2L, PPIA, SRP14, TMEM258, TSTD1 |
| 10 | CD74, CKB, COX7B, CPE, DHRS7, EEF2, HLA–DRA, PDLIM5, SELENOW, TXN, UBL5, UQCRB |
| 11 | ACTB, ATP6V0B, CD81, CD9, FLNA, FTH1, H3F3A, HNRNPA1, IGLC2, MYL9, NR4A1, SNHG25 |
| 12 | ATP5MPL, CD63, COX5B, FTL, MGP, NDUFB1, NDUFB11, PFN1, PTMA, SEC61G, SNHG8, SNRPD2 |
| 13 | ATP5MD, C12orf57, COMMD6, CSRP1, FASN, FXYD3, KLK3, MIF, NEAT1, SEC11C, SPINT2, VEGFA |
| 14 | CD99, CLU, DBI, FBLN1, KRT8, MALAT1, NPY, PLA2G2A, SEC61B, SERF2, SPDEF, TAGLN |
| 15 | COX6A1, COX7A2, EGR1, FAU, H3F3B, MSMB, MYH11, NKX3–1, PFDN5, SLC45A3, TFF3, TIMP1 |
| 16 | CALM2, CHCHD2, CST3, DES, GAPDH, H2AFJ, HINT1, IGHA1, KLK2, SNHG19, SPON2, SSR4 |
| 17 | CALR, CFD, COX8A, DSTN, GPX4, KRT18, PABPC1, PARK7, SERP1, STEAP2, TSC22D3 |
| 18 | TMEM59 |
| 19 | VIM |
| 20 | EEF1A1 |
| 21 | KLK4 |
| 22 | DUSP1 |
| 23 | ITM2B |
| 24 | MINOS1 |
| 25 | UQCR10 |
| 26 | NPM1 |
| 27 | DDT |
| 28 | HLA–B |
| 29 | COX6C |
| 30 | TRPM4 |
| 31 | FOS |

**Examples of biologically coherent modules.**   Several groups align with well-known PRAD biology:

- **Androgen–regulated luminal secretory program** (groups **13**, **15**, **8**, **5**, **16**, **21**): *KLK3/KLK2*, *NKX3–1*, *HOXB13*, *TMPRSS2*, *SLC45A3*, *MSMB*, *TFF3*, *SPDEF*. These genes are lineage markers or direct androgen receptor targets in prostate epithelium Cleutjens & et al. (1997); Wang & et al. (2009); Ewing & et al. (2012); Tomlins & et al. (2005).

- **Stromal smooth muscle and CAF–like ECM** (groups **7**, **14**, **15**): *ACTA2*, *CNN1*, *MYH11*, *TAGLN*, *FBLN1*, *DES* mark prostate stroma and myofibroblasts Chen & et al. (2021).

- **Antigen presentation and immunoglobulin** (groups **10**, **28**, **11**, **8**, **16**): *HLA–DRA*/*HLA–B*, *CD74*, *IGKC*/*IGLC2*/*IGHA1* reflect MHC and B–cell modules Collins et al. (1984).

- **Prostate–enriched antigens and secreted factors** (groups **7**, **17**, **16**): *AZGP1*, *STEAP2*, *SPON2* are well–documented prostate–enriched proteins with diagnostic or biological relevance Hubert & et al. (1999); Zhang & et al. (2012).

## C.5  ccRCC Dataset Selected Genes

The following genes were selected for the ccRCC (clear cell Renal Cell Carcinoma) dataset:

A2M, ACTA2, ACTB, ACTR3, ADIRF, AEBP1, AHNAK, ANGPTL4, ANPEP, ANXA2, APOC1, APOE, APOL1, APP, ARF1, ARF4, ARPC1B, ARPC2, ASPH, ATP1A1, ATP1B1, ATP5F1B, ATP5MC2, ATP5ME, B2M, BGN, BIRC3, BRI3, BSG, BST2, C19orf33, C1QA, C1QB, C1QC, C1R, C1S, C3, CA12, CALD1, CANX, CAV1, CCDC91, CCN1, CCN2, CCNI, CD24, CD44, CD63, CD68, CD74, CD81, CD99, CEBPD, CHCHD2, CIRBP, CLU, COL18A1, COL1A1, COL1A2, COL3A1, COL4A1, COL4A2, COL6A1, COL6A2, COL6A3, COX6C, CP, CPE, CRYAB, CST3, CSTB, CTSA, CTSB, CTSD, CTSZ, CXCR4, CYB5A, CYB5R3, DCN, DDIT4, DDX17, DEPP1, DUSP1, EEF1G, EIF1, EIF4A1, EIF4A2, EIF4G2, EIF4H, ENPP3, FCGRT, FGB, FKBP5, FLNA, FN1, FOS, FTH1, FTL, FXYD2, GABARAP, GLUL, GPX3, GSN, GSTP1, H3F3B, HINT1, HIST1H4C, HLA-A, HLA-DPA1, HLA-DPB1, HLA-DQA1, HLA-DQB1, HLA-DRA, HLA-DRB1, HLA-F, HMGB1, HNRNPA2B1, HNRNPA3, HPCAL1, HSP90AA1, HSP90B1, HSPA1B, HSPA5, HSPA8, HSPD1, HSPG2, HTRA1, IFI27, IFI30, IFI6, IFITM2, IGFBP3, IGFBP4, IGFBP5, IGFBP7, IGHA1, IGHG1, IGHG2, IGHG3, IGHG4, IGHM, IGKC, IGLC1, IL32, ITGA3, ITGB1, JCHAIN, KRT18, KRT8, LAPTM5, LGALS1, LGALS3, LGALS3BP, LY6E, LYZ, MCL1, MGP, MIF, MMP7, MYH9, MYL9, NCL, NDRG1, NDUFA4L2, NME2, NOP53, NPC2, NUPR1, P4HB, PARK7, PCBP1, PCBP2, PDIA6, PDK4, PDZK1IP1, PEBP1, PFDN5, PFKP, PGAM1, PGF, PLEC, PLIN2, PLOD2, PLTP, POSTN, PPDPF, PPP2CB, PRDX6, PRR13, PSAP, PTMA, PTMS, PTTG1IP, RARRES2, RASSF4, RGS5, RHOB, RNASET2, RPN2, S100A11, S100A6, SAT1, SCD, SEC61G, SELENOP, SERPINA1, SERPINE1, SERPING1, SNX3, SOD2, SPARC, SPINK13, SPP1, SQSTM1, SRRM2, SRSF2, SSR4, TAGLN, TAGLN2, TGFBI, TGM2, THBS1, TIMP1, TIMP3, TMBIM6, TMEM176A, TMEM176B, TMSB4X, TOMM7, TPM1, TPM2, TRAM1, TSC22D3, TUBA1B, TXN, TXNIP, TYROBP, UBA52, UBC, UQCRQ, VEGFA, VIM, VWF.

We clustered the 200 genes into 32 groups. The group membership is listed below.

| Group | Members |
| --- | --- |
| 0 | ANGPTL4, APP, ATP1B1, BGN, BIRC3, CAV1, CD74, FTH1, FTL, GABARAP, HNRNPA3, PTMA |
| 1 | GLUL |
| 2 | CYB5A |
| 3 | ARPC1B, C1R, CA12, CD63, CIRBP, COL4A2, DDIT4, FGB, HSP90B1, HTRA1, PDZK1IP1, RNASET2 |
| 4 | AEBP1, AHNAK, FKBP5, GSTP1, HIST1H4C, HLA–DQB1, HMGB1, HSPG2, IGFBP4, ITGB1, NDUFA4L2, PSAP |
| 5 | BST2, HLA–DQA1, HLA–DRA, HNRNPA2B1 |
| 6 | NUPR1 |
| 7 | KRT18 |
| 8 | CCNI |
| 9 | COL4A1 |
| 10 | C1QB |
| 11 | IGKC |
| 12 | ANPEP, COX6C, HSPA8, IFI30, IGHG4, IGHM, LY6E, MCL1, MGP, MYL9, NCL, PARK7 |
| 13 | APOC1, GSN, HSPD1, IFITM2, KRT8, LAPTM5, LGALS1, LGALS3, MMP7, MYH9, PLIN2, PTMS |
| 14 | ADIRF, ARF4, CCDC91, COL18A1, CTSZ, FXYD2, HLA–A, HSP90AA1, ITGA3, LYZ, NPC2, PEBP1 |
| 15 | APOE, ATP5MC2, BRI3, BSG, C1S, CP, DDX17, DEPP1, GPX3, IGFBP3, LGALS3BP, PGAM1 |
| 16 | ACTB, ARPC2, C1QA, C1QC, CCN1, COL6A3, CYB5R3, IGFBP5, MIF, PFDN5, PPDPF, S100A11 |
| 17 | A2M, ACTR3, ANXA2, APOL1, ATP1A1, HLA–DPA1, HSPA1B, HSPA5, NDRG1, PLTP, RAR-RES2, RGS5 |
| 18 | ATP5F1B, CD81, CST3, CXCR4, EIF4H, ENPP3, FCGRT, FN1, IGHG2, NOP53, PRR13, RPN2 |
| 19 | CALD1, CTSA, CTSB, IFI6, IGHA1, IGHG3, NME2, PCBP2, PDIA6, POSTN, PRDX6, RHOB |

| 20 | ARF1, CCN2, CD24, CD44, COL6A2, EIF4G2, IFI27, IGHG1, JCHAIN, P4HB, PGF, PLOD2 |
| 21 | ACTA2, ASPH, C3, CANX, CD99, CEBPD, COL1A1, DCN, EIF4A1, FOS, HLA–DPB1, HP-CAL1 |
| 22 | ATP5ME, CLU, CPE, CSTB, DUSP1, EEF1G, EIF1, FLNA, HLA–F, IGLC1, PLEC, RASSF4 |
| 23 | C19orf33, CD68, CHCHD2, COL1A2, COL3A1, COL6A1, CRYAB, H3F3B, HINT1, IL32, PDK4, PPP2CB |
| 24 | CTSD |
| 25 | PTTG1IP |
| 26 | EIF4A2 |
| 27 | PFKP |
| 28 | HLA–DRB1 |
| 29 | IGFBP7 |
| 30 | B2M |
| 31 | PCBP1 |

**Examples of biologically coherent modules (by groups).** Several groups align with well-known ccRCC or tumor-microenvironment programs:

- **Hypoxia/lipid metabolism and ECM remodeling**: groups **3**, **4**, **15**, **18**, **20**. Representative genes include *NDUFA4L2*, *PLIN2*, *CA12*, *ENPP3*, *PLOD2*, *PGF*, *FN1*. These are classic HIF–hypoxia targets or matrix/secretory programs frequently upregulated in ccRCC Kubala et al. (2023); Cao et al. (2018); Ivanov et al. (2001); Thompson et al. (2018).

- **Antigen presentation and interferon-stimulated genes**: groups **5**, **14**, **21**, **28**. Genes such as *HLA–DRA/DRB1/DPA1/DPB1*, *HLA–A*, *BST2*, *IFI27*, *IFI6*, *IFITM2*, *LY6E* mark MHC-II antigen processing and IFN response Collins et al. (1984); Ortega-Prieto et al. (2024).

- **C1Q⁺ macrophage module**: groups **10**, **16**, **23** with *C1QA/B/C*, *CD68*, often alongside *APOE/LGALS3* Lindblom et al. (2003); He et al. (2016b).

- **Pericyte/smooth-muscle and stromal ECM**: groups **19**, **21**, **23** with *ACTA2*, *MYL9*, *CALD1*, *RGS5*, and collagens *COL1A1/1A2/3A1/6A1/6A3*, *DCN*, *POSTN*, *FN1* Lindblom et al. (2003); He et al. (2016b).

- **Plasma-cell immunoglobulins**: groups **11**, **12**, **18**, **19**, **20** featuring *IGHG1/2/3/4*, *IGHM*, *IGHA1*, *IGKC*, *JCHAIN* Xu et al. (2020); Onieva et al. (2022).

# D   IMPLEMENTATION DETAILS AND REPRODUCIBILITY

**Reproducibility** To ensure consistent results, we fix the random seed at 2021 (configurable via `--seed`). The `fix_seed` function controls randomness in Python, NumPy, PyTorch, and CUDA operations. For complete reproducibility, set Lightning's `deterministic=True` and apply cuDNN flags as recommended in PyTorch documentation. All inference scripts use fixed seeds.

**Training Setup** We use a learning rate of 1e-4 with weight decay of 1e-4 and gradient clipping at 1.0. The learning rate scheduler is disabled by default. Training uses batch size 256 while validation and testing use batch size 64, with 4 data loading workers and `pin_memory=True`.

The model uses 768-dimensional embeddings with 8 transformer layers, 8 attention heads, and MLP ratio of 3.0. Dropout is set to 0.0 for training and 0.1 for evaluation. Multi-scale patches are configured as `gene_patch_nums = (1, 4, 8, 40, 100, 200)` with a vocabulary size of `max_gene_count + 1` (default 2000). All models train for 50 epochs with early stopping.

**Heteroscedastic Gaussian head.** At the final scale, we parameterize the Gaussian loss as $\sigma^2 = \alpha\mu + \beta$ with fixed $\alpha = 0.01$ and $\beta = 1.0$ for all experiments; we observed that the method is reasonably robust to moderate changes of these values.

**Multi-GPU Training** We use DDP for multi-GPU training with `accumulate_grad_batches = 1` and `find_unused_parameters = False`. DDP is automatically enabled when multiple GPUs are detected.

# E    ADDITIONAL VISUALIZATION RESULTS

To provide comprehensive spatial visualization comparisons across different genes, we present additional visualization results on the HER2ST dataset using the SPA148 sample. These visualizations demonstrate the spatial expression patterns predicted by our GenAR method compared to baseline approaches across a diverse set of genes with different expression characteristics and biological functions. The selected genes represent various functional categories including structural proteins, growth factors, immune-related genes, and metabolic enzymes, showcasing the generalization capability of our method across different gene types.

# F    LARGE LANGUAGE MODEL USAGE

Large Language Models were used as general-purpose writing assistance tools to improve the grammar, clarity, and organisation of the manuscript. The core research contributions, methodology, experimental design, and scientific insights are entirely original work by the authors.

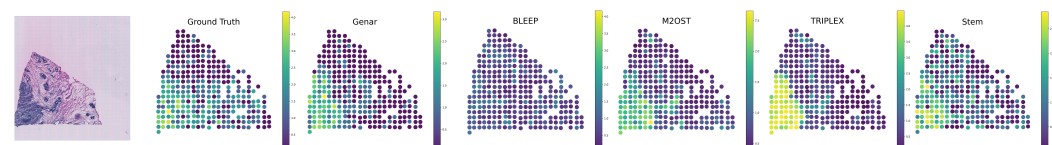

Figure 5: Spatial visualization comparison of C12orf57 gene expression prediction on HER2ST SPA148 sample.

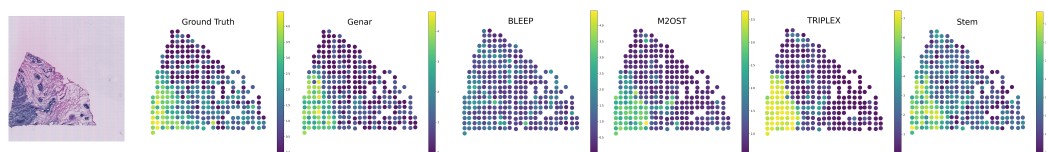

Figure 6: Spatial visualization comparison of EIF4G1 gene expression prediction on HER2ST SPA148 sample.

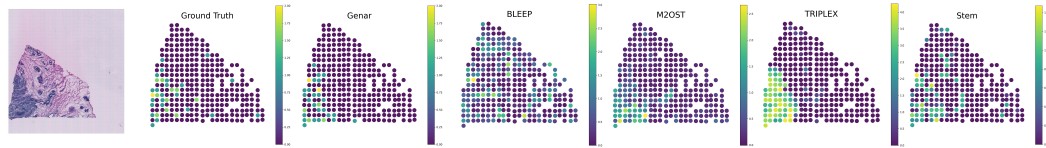

Figure 7: Spatial visualization comparison of FNBP1L gene expression prediction on HER2ST SPA148 sample.

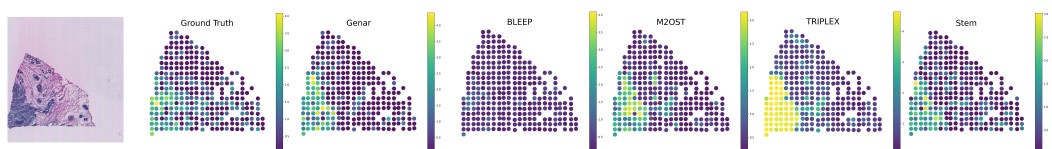

Figure 8: Spatial visualization comparison of IGFBP2 gene expression prediction on HER2ST SPA148 sample.

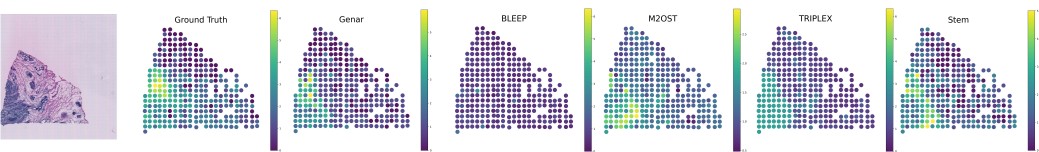

Figure 9: Spatial visualization comparison of ISG15 gene expression prediction on HER2ST SPA148 sample.

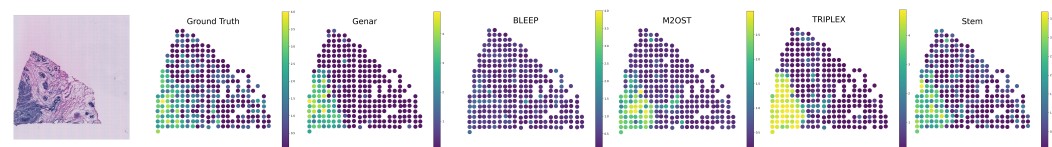

Figure 10: Spatial visualization comparison of NUCKS1 gene expression prediction on HER2ST SPA148 sample.

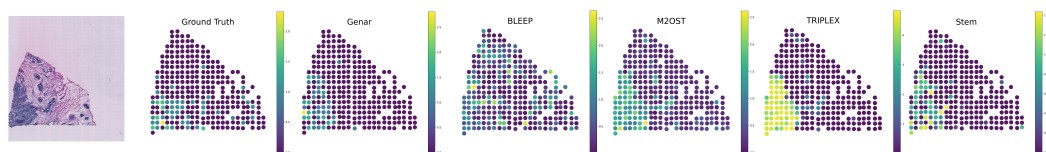

Figure 11: Spatial visualization comparison of ORMDL3 gene expression prediction on HER2ST SPA148 sample.

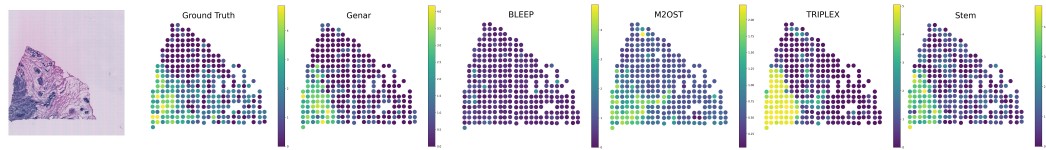

Figure 12: Spatial visualization comparison of PPP1R1B gene expression prediction on HER2ST SPA148 sample.

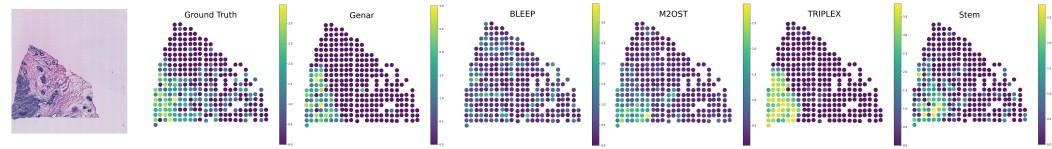

Figure 13: Spatial visualization comparison of SF3B5 gene expression prediction on HER2ST SPA148 sample.

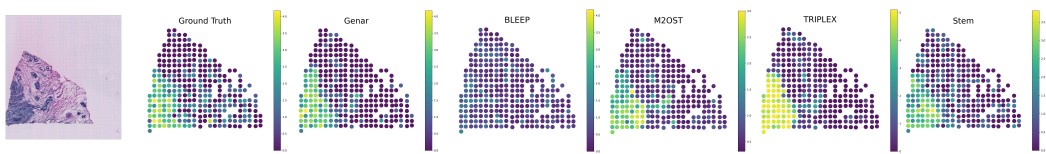

Figure 14: Spatial visualization comparison of SSR2 gene expression prediction on HER2ST SPA148 sample.

