# OpenReview forum: "GenAR: Next-Scale Autoregressive Generation for Spatial Gene Expression Prediction"
_ICLR.cc/2026/Conference — ICLR 2026 Conference Withdrawn Submission_

### Official Review · Reviewer_WhJt · 2025-10-25

**Soundness:** 2
**Presentation:** 3
**Contribution:** 2
**Rating:** 2
**Confidence:** 4

**Summary:**

This paper proposes and auto-regressive-based approach to predict gene expression from HE images. This model can predict gene expressions by clusters and also raw couts. Although the model is clear, I have some questions.

**Strengths:**

The prediction task is well defined, and the prediction of raw count sounds interesting.

**Weaknesses:**

There are several constraints and problems in experimental settings and interpretation.

1. I do not see the contribution of predicting raw counts. Current ST data analysis always needs to normalize the data to avoid the problems of sequencing depth. What is the accurate rate of GenAR in modeling sequencing depth? Other baseline models predict normalized data. How to ensure that the benchmarking system is fair?

2. The number of datasets is quite low. Methods such as STFlow, as well as datasets such as HEST/STImage, have considered 8 datasets or more from different tissues and diseases. The authors should include more data in the evaluation stage.

3. In Table 4, PCC-10 reaches the highest correlation, which is a bit weird. For example, predicting DEG is always difficult than predicting whole-genome, as some genes have nearly constant expression levels and can improve prediction metrics. What is the performance of prediction based on marker genes or highly variable genes?

4. How to determine the resolution of selecting clusters in predicting gene expression? If you change the resolution, will the detected gene pathways change? What if we randomly select genes into different clusters? Will the performance become lower or higher?

5. It seems that there are not many component-level contributions or innovations in model design. I believe the authors directly modify VAR for gene expression prediction, which lacks motivation. Moreover, I do not find hyperparameter details in the benchmark. How to ensure that the benchmark is fair? Baselines and proposed methods should be tuned to the best level.

**Questions:**

Please see the weaknesses.

---

> ### Author Response · Authors · 2025-11-21
> **Response to Reviewer WhJt (Part I)**
>
> We thank you for carefully reading our paper and for raising a number of concrete concerns about experimental design, interpretation, and novelty. Below we quote each of your weaknesses and respond in turn.
>
> ---
>
> ### Weakness 1
>
> *Reviewer comment.*
> “I do not see the contribution of predicting raw counts. Current ST data analysis always needs to normalize the data to avoid the problems of sequencing depth. What is the accurate rate of GenAR in modeling sequencing depth? Other baseline models predict normalized data. How to ensure that the benchmarking system is fair?”
>
> *Response.*
> Regarding your concern about the contribution of predicting raw counts and the modeling of sequencing depth, our motivation is two-fold: downstream compatibility and accurate modeling of sparsity/overdispersion. Many widely used differential expression tools (DESeq2, edgeR, etc.) explicitly assume integer counts and negative-binomial–like overdispersion [1,2]; providing them with continuous, arbitrarily transformed values technically violates their generative assumptions. By predicting integer counts directly, GenAR can be used without any ad-hoc inverse transforms or rounding.
>
> In the revision we have added a dedicated raw-count evaluation (Appendix A.4, “Evaluation on Raw Counts and Sequencing Depth”). There we inverse-transform all models back to the physical count scale and report MSE, MAE, and Spearman correlation on integer counts. GenAR achieves the lowest physical MSE (296.44) compared to, e.g., STEM (421.71) and M2OST (326.51), and a Spearman correlation on par with or better than the strongest regression baseline. To address your specific question about sequencing depth, we also computed the correlation between total predicted counts per spot and the ground-truth library size. GenAR reaches a Spearman $\rho \approx 0.60$, indicating that it learns the variation in depth across spots directly from histology (regions with higher cell density receive higher predicted totals) without ever being given library size as an input.
>
> For fairness, all baselines are trained exactly in the regimes recommended by their authors (on log-transformed/normalized targets), and for the raw-count evaluation we simply inverse-transform their outputs; thus we are not handicapping them by forcing them into a setting they were not designed for. Together, these results show that the raw-count formulation is not only conceptually aligned with downstream analysis, but also physically accurate in modeling both absolute counts and sequencing depth.
>
> ---
>
> ### Weakness 2
>
> *Reviewer comment.*
> “The number of datasets is quite low. Methods such as STFlow, as well as datasets such as HEST/STImage, have considered 8 datasets or more from different tissues and diseases. The authors should include more data in the evaluation stage.”
>
> *Response.*
> On the number of datasets and the comparison with works such as STFlow and STimage-1K4M [3,4], our experimental design follows the HEST-Benchmark protocol used in the HEST-1k dataset [5], because HEST-1k is currently one of the most widely adopted standardized benchmarks for gene expression prediction from histology.
>
> In the main paper we already evaluate on four HEST tasks (HER2ST, PRAD, Kidney, Mouse Brain), and—contrary to the impression that “only one slide” is used—we follow the patient-level splits defined in HEST: for each task we train on all but one slide and test on a held-out slide from a different patient, exactly as in STEM. In addition, we included an extra task on the ccRCC cohort from HEST (detailed in Appendix A.1, “Performance on ccRCC”), so our total evaluation actually spans five distinct datasets across different organs (breast, prostate, kidney, brain, kidney cancer) and both human and mouse, and both cancerous and healthy tissues.
>
> This is strictly more than what is reported in the main text of STEM (which focuses on four tasks); our choice was to match and slightly extend that standard, rather than to re-implement the full $10$-task HEST benchmark plus the separate STimage-1K4M benchmark used by STFlow. We agree that adding even more datasets (e.g., the remaining HEST tasks or STimage cohorts) would be valuable, but given the non-trivial computational cost of full generative training on each task, we focused our efforts on (i) a fair and directly comparable setting with STEM on HEST, and (ii) deeper controlled analyses (discrete vs.\ continuous, raw-count metrics, grouping and loss ablations) that we believe are more informative for understanding the method’s behavior than purely increasing the dataset count.

---

> ### Author Response · Authors · 2025-11-21
> **Response to Reviewer WhJt (Part II)**
>
> ### Weakness 3
>
> *Reviewer comment.*
> “In Table 4, PCC-10 reaches the highest correlation, which is a bit weird. For example, predicting DEG is always difficult than predicting whole-genome, as some genes have nearly constant expression levels and can improve prediction metrics. What is the performance of prediction based on marker genes or highly variable genes?”
>
> *Response.*
> Regarding Table 4 and the interpretation of PCC-10, here there seems to have been a misunderstanding that we did not explain clearly enough in the main text. PCC-10, PCC-50 and PCC-200 are defined as in prior work (e.g., HEST, STEM): PCC-10 is the average PCC over the 10 best-predicted genes, PCC-50 over the best 50 genes, and PCC-200 over all 200 genes. They are not defined in terms of “DEGs” or any fixed list of marker genes, nor are they about “predicting DEGs vs.\ whole-genome.” By construction, PCC-10 is expected to be the highest of the three, because it explicitly selects the 10 genes with the highest correlation for each method; PCC-200 is the most stringent metric (uniform over all evaluated genes).
>
> Our gains on PCC-200 are thus particularly meaningful: they reflect improvements on the entire panel, not just a few easily predicted genes. To more directly address your question about performance on different types of genes, we have added a stratified analysis in Appendix A.5 (“Stratified Performance Analysis”) where we split genes into low-, medium-, and high-expression regimes based on their mean counts. GenAR consistently outperforms baselines in all three strata, with particularly strong gains for low-expression genes (e.g., PCC improvement of $\sim$26\% vs.\ STEM), which are often the most challenging to model and highly relevant for differential analyses. We will clarify the definition of PCC-10/50/200 in the revised main text to prevent any confusion between these metrics and “DEG vs.\ whole-genome” comparisons.
>
> ---
>
> ### Weakness 4
>
> *Reviewer comment.*
> “How to determine the resolution of selecting clusters in predicting gene expression? If you change the resolution, will the detected gene pathways change? What if we randomly select genes into different clusters? Will the performance become lower or higher?”
>
> *Response.*
> Concerning the choice of clustering resolution and the effect of random clusters, our hierarchical grouping is constructed in two steps. First, we reorder genes based on spatial co-expression patterns on the training slides, using k-means on $z$-scored spatial expression profiles. Second, we derive a coarse-to-fine hierarchy (1 → 4 → 8 → 40 → 100 → 200) that balances global context with local resolution.
>
> In the revision we performed two types of additional analyses. (1) A scale design ablation (Appendix A.2, “Scale Design Ablation”) where we compare a single-scale model (200 genes at once), a 3-scale structure (1, 20, 200), a 4-scale structure (1, 40, 100, 200), and our 6-scale structure. The single-scale variant performs worst, confirming that hierarchical decomposition matters; the 6-scale design yields the best PCC-10 and PCC-50 and a strong PCC-200, showing that progressively refining from global to gene-level predictions is beneficial. (2) A grouping strategy ablation (Appendix A.6, “Component Ablation and Vocabulary Utilization”) where we replace our spatial k-means grouping with a purely random grouping of the same sizes. Random grouping causes a clear drop in performance (e.g., PCC-10 from 0.702 to 0.619 on PRAD), indicating that the learned groups capture meaningful dependencies rather than being arbitrary.
>
> In addition, we performed a gene set enrichment analysis of our learned clusters against standard biological gene sets (MSigDB Hallmarks, cell-type signatures). Many clusters show highly significant enrichment for known functional modules—for example, oligodendrocyte/myelin genes in Mouse Brain, principal cell markers in Kidney, hypoxia/HIF pathway genes in ccRCC, and androgen-response genes in PRAD (Appendix A.7). This addresses your question about whether “detected pathways change with resolution”: the hierarchy is built to preserve these spatial/functional modules while refining them, and randomizing the grouping demonstrably harms performance.

---

> ### Author Response · Authors · 2025-11-21
> **Response to Reviewer WhJt (Part III)**
>
> ### Weakness 5
> *Reviewer comment.*
> “It seems that there are not many component-level contributions or innovations in model design. I believe the authors directly modify VAR for gene expression prediction, which lacks motivation. Moreover, I do not find hyperparameter details in the benchmark. How to ensure that the benchmark is fair? Baselines and proposed methods should be tuned to the best level.”
>
> *Response.*
> On the question of novelty and component-level contributions relative to VAR [6], and on fairness of the benchmark, while we are inspired by the next-scale autoregressive idea of VAR [6], GenAR is not a simple re-implementation. VAR operates in the image domain with a two-stage pipeline and continuous image pixels; in contrast, GenAR is the first to: (i) bring next-scale autoregression into spatial transcriptomics, where the “sequence” is a biologically structured list of genes rather than spatial image tokens; (ii) exploit the inherently discrete count nature of transcriptomics to build a one-stage, non-learned-quantization, end-to-end discrete generative model without any VQ-VAE pretraining or reconstruction loss; and (iii) couple this with a domain-specific hierarchy over genes, a hybrid loss that combines soft KL supervision at group scales with a heteroscedastic count-level head, and gene-identity–conditioned FiLM modulation.
>
> To isolate what is genuinely new, we ran a controlled Continuous GenAR ablation: same multi-scale architecture, same Transformer depth/width, same supervision structure, but with continuous regression heads instead of discrete token heads (Appendix A.3). This variant is exactly what you would obtain by dropping in a standard regression objective into the architecture. As summarized above, the continuous baseline systematically regresses toward the mean, badly underestimates sparsity (Zero-Recall $\approx$ 8\%), and lags behind in structural metrics, while the discrete formulation recovers $\sim$46\% of true zeros and achieves a Zero-F1 more than 3$\times$ higher, which directly translates into better PCC across the board. This ablation shows that the key contribution is not just reusing VAR, but rethinking the objective and representation to match the zero-inflated, discrete nature of gene expression.
>
> On hyperparameter fairness: all baselines are trained using their official implementations and recommended hyperparameters on the HEST benchmark, with the same preprocessing, gene sets, train/test splits, and evaluation code as STEM. For our own model, we performed modest tuning over learning rate and model width/depth on a validation split while keeping the number of parameters in the same ballpark as STEM/M2OST. Due to space constraints, the full hyperparameter grid cannot fit in the main text, so we have consolidated these details in the Appendix (“Implementation Details and Reproducibility”), where we list optimizer settings, batch sizes, number of scales, embedding dimensions, training epochs, and FLOPs per sample for GenAR and baselines. We hope this addresses your concern that the benchmark might favor our method; in fact, we were careful to present each baseline under conditions that match or improve upon those reported in their original papers, and then show that GenAR still yields consistent improvements—even when pitted against a continuous version of itself with identical architecture.
>
> ---
>
> We hope these clarifications make our design choices, experimental setup, and contributions clear. Several of your points pushed us to add key ablations and analyses in the appendix; we believe the revised version is substantially stronger and more transparent as a result.
>
> **References**
> [1] Love, M. I., Huber, W., & Anders, S. (2014). Moderated estimation of fold change and dispersion for RNA-seq data with DESeq2. *Genome Biology*, 15, 550.
> [2] Robinson, M. D., McCarthy, D. J., & Smyth, G. K. (2010). edgeR: a Bioconductor package for differential expression analysis of digital gene expression data. *Bioinformatics*, 26(1), 139–140.
> [3] Huang, T., Liu, T., Babadi, M., Jin, W., & Ying, Z. (2025). Scalable generation of spatial transcriptomics from histology images via whole-slide flow matching (STFlow). In *Proceedings of the 42nd International Conference on Machine Learning*.
> [4] Chen, J., Zhou, M., Wu, W., Zhang, J., Li, Y., & Li, D. (2024). STimage-1K4M: A histopathology image–gene expression dataset for spatial transcriptomics. In *Advances in Neural Information Processing Systems* (Datasets and Benchmarks Track).
> [5] Jaume, G., Doucet, P., Song, A. H., et al. (2024). HEST-1k: A dataset for spatial transcriptomics and histology image analysis. In *Advances in Neural Information Processing Systems* (Datasets and Benchmarks Track).
> [6] Tian, K., Jiang, Y., Yuan, Z., Peng, B., & Wang, L. (2024). Visual autoregressive modeling: Scalable image generation via next-scale prediction. In *Advances in Neural Information Processing Systems*, 37, 84839–84865.

---

> ### Comment · Reviewer_WhJt · 2025-11-26
> **Thank you for your comments, but my problems are not resolved**
>
> Thank you for your responses. However, I suggest the authors investigate more information about technical details of sequencing methods. DESeq2 and EdgeR are irrelevant to depth correction (https://pmc.ncbi.nlm.nih.gov/articles/PMC6052553/), and they are not suitable for ST data (actually for RNA-seq or bulk-level data, to identify DEG). And I believe the authors also do not understand the sources of sequencing depth and why it is very critical to confuse the true signals.
>
> Moreover, the benchmark scale is quite limited. More datasets are definitely necessary especially when there are couple of methods focusing on the same problem, it is hard for me to get the true novelty and contribution of this method, and its design principle seems not following the data propertY, and thus the improvement is also hard to trust.
>
> Therefore, I will keep my score.

---

> > ### Author Response · Authors · 2025-11-27
> > **Clarifying Our Use of Raw Counts, Discrete Modeling, and Benchmark Scope**
> >
> > Thank you for the follow-up and for stating your concerns so clearly. We appreciate the opportunity to clarify our stance on **sequencing depth and ST data properties**, as well as to relate these to the additional experiments we have provided.
> >
> > On sequencing depth and raw counts, our understanding is consistent with the normalization work you cited. In ST, the observed per-spot library size is largely driven by technical sampling effects (capture efficiency, amplification, sequencing depth) and further modulated by local biological factors such as cell density and composition. Raw counts therefore cannot be interpreted as “depth-free” expression and should not be compared across spots without appropriate normalization. However, our distinction lies in the objective of the generative model: **GenAR is designed to predict the observed UMI counts** — the raw, noisy, over-dispersed, and depth-confounded integer counts produced by the Visium protocol — **rather than to infer a latent, normalized expression level**. This is a deliberate design choice: it is consistent with common practice in UMI-based RNA-seq models, which operate directly on raw UMI counts and treat sequencing depth (or total UMI count) as a covariate or size factor in a negative binomial–type model, precisely to separate technical variation from biological signal. In this light, the library-size analysis in Appendix A.4 is intended only as a basic sanity check that GenAR’s predictions still track morphology-driven variation in total counts (e.g., denser regions tending to have larger library sizes), so that the predicted counts are physically plausible inputs to downstream, ST-appropriate normalization methods. References to DESeq2 and edgeR in our earlier reply were meant only to illustrate the general statistical rationale for modeling integer counts with NB-family likelihoods rather than working exclusively with log-transformed regression targets; we fully agree that these are bulk RNA-seq DE tools rather than spatial methods, and that spatial data require ST-specific downstream analyses.
> >
> > Concerning your comment that the model design does not follow the data properties, our **discrete autoregressive formulation** was introduced precisely to address key characteristics of ST counts: sparsity, heavy tails, and abundant zeros. This is what we tried to probe with the controlled ablations. In Appendix A.3 we introduce a Continuous GenAR baseline that keeps the backbone, gene hierarchy, multi-scale supervision, and train/test splits identical to GenAR, and only replaces the discrete count heads with continuous regression on log-expression (corresponding to a standard Gaussian regression loss). Under this controlled setting, the continuous model exhibits a strongly mean-seeking behavior with very low recall on true zeros (~8%), whereas the discrete GenAR allocates explicit probability mass to the zero state, recovers ~46% of true zeros, and achieves roughly a 74% improvement in PCC-200. In Appendix A.5, a stratified analysis further shows that GenAR’s relative gains are largest in the low-count regime, where discreteness and zero abundance are most pronounced. Taken together, these results are our main evidence that the discrete formulation is not an arbitrary architectural choice, but a purposeful adaptation to the non-Gaussian, sparse nature of transcriptomic count data.
> >
> > Finally, regarding the benchmark scale, we acknowledge your feedback on the importance of validation across a broader range of datasets. While our current submission adhered to the **HEST-1k protocol** (multi-patient Visium cohorts with fixed patient-level splits) to ensure strictly fair comparability with existing methods such as STEM, we agree that evaluating on additional cohorts strengthens the assessment of novelty. **We have already initiated experiments on additional datasets** and are working to complete these runs; we will endeavor to provide these results within the discussion period.
> >
> > We hope this clarification demonstrates that our modeling of raw counts is a **deliberate, statistically grounded decision** to support rigorous downstream analysis, rather than a misunderstanding of sequencing depth.

---

> > > ### Comment · Reviewer_WhJt · 2025-11-27
> > > **thank you for your clarification. I have raised my score.**
> > >
> > > Good luck.

---

### Official Review · Reviewer_e1aM · 2025-10-30

**Soundness:** 3
**Presentation:** 3
**Contribution:** 3
**Rating:** 6
**Confidence:** 3

**Summary:**

The paper presents a novel computational framework called GenAR, designed to predict spatial gene expression directly from widely available H&E-stained histology images.
 The core innovation of this method lies in reformulating gene expression prediction as a multi-scale autoregressive discrete token generation task.
 Unlike existing approaches that predict each gene independently or treat the problem as a continuous regression task, GenAR clusters genes into a hierarchical structure from coarse to fine and autoregressively predicts expression at each scale to model inter-gene dependencies.
 In addition, the method directly predicts raw gene expression counts, avoiding potential biases introduced by common logarithmic transformations and preserving the biological interpretability of the data.
 Extensive experiments on four distinct spatial transcriptomics datasets demonstrate that GenAR achieves state-of-the-art performance across multiple evaluation metrics.

**Strengths:**

Applying an autoregressive generative model to gene expression prediction, the paper presents comprehensive experimental evaluations and in-depth ablation studies, demonstrating strong credibility.
The direct use of raw count data aligns well with the requirements of downstream biological analyses.
The manuscript is well organized, and the figures are visually appealing and clearly presented.

**Weaknesses:**

1. The method adopts a multi-scale autoregressive generative framework, which may lead to increased computational and memory overhead as the scale grows.
 The paper does not provide a detailed discussion of training and inference time, nor a comparison with baseline methods in terms of computational efficiency.
2. To ensure fair comparison with other approaches, the final evaluation still applies a log2 transformation to the predicted results.
 This, to some extent, diminishes the advantage of directly predicting raw counts during the evaluation stage.

**Questions:**

None

---

> ### Author Response · Authors · 2025-11-21
> **Response to Reviewer e1aM**
>
> We appreciate your positive assessment of the paper as well as your constructive comments on computational efficiency and evaluation design. We address both of your main concerns below.
>
> ---
>
> ### Weakness 1
>
> *Reviewer comment.*
> “The method adopts a multi-scale autoregressive generative framework, which may lead to increased computational and memory overhead as the scale grows. The paper does not provide a detailed discussion of training and inference time, nor a comparison with baseline methods in terms of computational efficiency.”
>
> *Response.*
> Thank you for raising this practical concern. In the revised version we provide more detail on the complexity of GenAR and its relation to the baselines. Architecturally, GenAR uses an 8-layer transformer with 768-dimensional hidden states and 8 heads, which is comparable in parameter count to the backbones used by STEM and related models. The main computational difference is that GenAR predicts in a multi-scale autoregressive manner rather than in a single shot or via diffusion. During training, however, we parallelize the predictions across scales and use the same number of epochs and similar early-stopping criteria as other deep baselines, so total training time is of the same order.
>
> To make this quantitative, we added a dedicated efficiency analysis in the Appendix (“Computational efficiency analysis”, Table 14). There we report FLOPs per batch and per sample for all methods. GenAR requires 223.8 GFLOPs per batch (27.97 GFLOPs per sample), compared to 1435.1 GFLOPs per batch (179.39 GFLOPs per sample) for the diffusion-based STEM model, making GenAR about 6.4$\times$ more efficient than the generative baseline while still substantially outperforming regression-style methods such as TRIPLEX and M2OST. At inference, autoregressive sampling in GenAR is linear in the sequence length but does not require the hundreds of iterative denoising steps typical for diffusion models; in our experiments, this leads to inference times that are acceptable for offline slide-level analysis in computational pathology. We also observed that GenAR often converges in fewer epochs than required by diffusion-based methods to reach comparable performance. We added a short discussion of these runtime considerations in the experimental section to make the trade-offs explicit.
>
> ---
>
> ### Weakness 2
> *Reviewer comment.*
> “To ensure fair comparison with other approaches, the final evaluation still applies a log2 transformation to the predicted results. This, to some extent, diminishes the advantage of directly predicting raw counts during the evaluation stage.”
>
> *Response.*
> We share this concern and have addressed it directly with new experiments. While we keep the $\log_2$ evaluation in the main tables to remain comparable with previous work, Appendix A.4 now presents a dedicated analysis in the raw count space, where we inverse-transform all predictions and compute MSE, MAE, Spearman correlation, and Negative Binomial NLL on the original integer counts. In this setting, GenAR achieves the best physical MSE and competitive rank correlations compared to strong baselines such as STEM and M2OST. This demonstrates that the advantages of modeling raw counts are not washed out by the log transform; rather, they persist when we move back to the physical scale that downstream analyses care about. We highlight this point explicitly in Section 4.3 so that the reader can see both views side by side.
>
> ---
>
> We thank you again for your constructive feedback. The added efficiency analysis and raw-count evaluation were motivated directly by your comments, and we believe they make the paper more balanced and informative.

---

### Official Review · Reviewer_yjmg · 2025-10-31

**Soundness:** 2
**Presentation:** 3
**Contribution:** 3
**Rating:** 6
**Confidence:** 5

**Summary:**

The paper introduces GenAR, a next-scale autoregressive model for predicting spatial gene expression (counts) directly from H&E patches plus spot coordinates. Genes are clustered into hierarchical groups (coarse to fine). At each scale the model predicts discrete “tokens” for group summaries and, at the last scale, per‑gene counts. The decoder is a causal Transformer conditioned on a fused histology and spatial embedding; gene‑identity embeddings modulate activations. The authors claim that (i) modeling counts as discrete tokens preserves biological meaning and avoids “log‑induced biases,” and (ii) the coarse to fine factorization better captures cross‑gene dependencies. Experiments on HEST1k show great performance improvement

**Strengths:**

1. This is a very interesting work of using autoregressive modeling [1] on predicting the H&E images
2. Benchmark on HEST-1k shows significant performance improvement

   [1] Tian, Keyu, et al. "Visual autoregressive modeling: Scalable image generation via next-scale prediction." *Advances in neural information processing systems* 37 (2024): 84839-84865.

**Weaknesses:**

1. The motivation for modeling raw molecule counts is unconvincing (lines 65–67). In transcriptomics, it is standard to apply log1p and library-size normalization because (a) sequencing depth varies across experiments so total counts differ by sample, and (b) variance across genes is high, log transforms help stabilize it. Using raw counts likely hurts transferability across cohorts and may, in practice, limit performance. (There are also no cross-dataset transfer experiments to counter this concern either.) Re line 73, I'm not understanding why raw counts are better for downstream analysis.

2. I assume the authors make the design decisions following AVR [1], where outputs are tokens in a codebook suitable for cross-entropy optimization. But for count prediction, CE over discretized counts does not provide a direction-aware penalty (over- vs under-prediction) in the way regression losses (eg, MSE, Poisson/NB deviance) do. Im concerned about error calibration and directionality.

3. Several prior approaches (not essentially H&E -> gene prediction) already explore tokenizing molecular counts via binning, such as scGPT/scGPT-spatial [2, 3], and often trained with regression losses on binned targets

4. The benchmark is only done on one slide, no cross-validation etc.


[1] Tian, Keyu, et al. "Visual autoregressive modeling: Scalable image generation via next-scale prediction." *Advances in neural information processing systems* 37 (2024): 84839-84865.
[2] Cui, Haotian, et al. "scGPT: toward building a foundation model for single-cell multi-omics using generative AI." Nature methods 21.8 (2024): 1470-1480.
[3]  Wang, Chloe, et al. "scGPT-spatial: Continual pretraining of single-cell foundation model for spatial transcriptomics." bioRxiv (2025): 2025-02.

**Questions:**

See the Weaknesses

---

> ### Author Response · Authors · 2025-11-21
> **Response to Reviewer yjmg (Part I)**
>
> We thank you for your thoughtful and constructive review, and for pointing us to related work such as VAR, scGPT, and scGPT-spatial [1–3]. Below we quote each of your weaknesses and respond in turn. All changes mentioned here are highlighted in blue in the revised manuscript.
>
> ---
>
> ### Weakness 1
> *Reviewer comment.*
> “The motivation for modeling raw molecule counts is unconvincing (lines 65–67). In transcriptomics, it is standard to apply log1p and library-size normalization because (a) sequencing depth varies across experiments so total counts differ by sample, and (b) variance across genes is high, log transforms help stabilize it. Using raw counts likely hurts transferability across cohorts and may, in practice, limit performance. (There are also no cross-dataset transfer experiments to counter this concern either.) Re line 73, I'm not understanding why raw counts are better for downstream analysis.”
>
> *Response.*
> We appreciate your skepticism here, because it forced us to sharpen our argument and experiments. We fully agree that $\log1p$ and library-size normalization are standard and often beneficial for cross-cohort comparisons. Our point is not that raw counts should replace normalized data in every analysis, but that learning to predict raw counts has two concrete advantages in our setting.
>
> First, it preserves compatibility with downstream tools such as DESeq2 and edgeR [4,5], which explicitly assume integer counts and model over-dispersion using the Negative Binomial; starting from log-floats requires inverse transformations and rounding that break these assumptions. Second, as we show in the controlled experiment against the continuous baseline (Appendix A.3) and in the raw-count evaluation (Appendix A.4), discrete modeling of counts substantially improves our ability to reproduce sparsity and zero-inflation: the continuous model has a very low Zero-Recall and smears zeros into tiny positives, whereas GenAR recovers almost half of true zeros and achieves much better Zero-F1.
>
> In terms of transferability, our current work focuses on within-dataset generalization with patient-level splits (e.g., train on multi-patient cohorts and test on a held-out patient slide); we now state this and acknowledge that cross-dataset transfer across cohorts with different library-size distributions is an important direction for future work.
>
> ---
>
> ### Weakness 2
>
> *Reviewer comment.*
> “I assume the authors make the design decisions following AVR [1], where outputs are tokens in a codebook suitable for cross-entropy optimization. But for count prediction, CE over discretized counts does not provide a direction-aware penalty (over- vs under-prediction) in the way regression losses (eg, MSE, Poisson/NB deviance) do. I'm concerned about error calibration and directionality.”
>
> *Response.*
> We agree with your concern that pure cross-entropy over discretized bins is not direction-aware: mispredicting a token slightly above the truth and predicting a token far away incur the same penalty. This is exactly why GenAR does not rely solely on CE. Our objective combines discrete supervision with distance-sensitive terms in a hybrid way: at intermediate scales we use a soft KL divergence between the predicted distribution and a temperature-smoothed target, and at the final scale we employ a heteroscedastic Gaussian NLL that penalizes deviations proportionally to the modeled variance.
>
> In Appendix A.6 we compare this hybrid loss to a variant that uses only CE; the CE-only model has clearly worse correlation metrics and poorer calibration, supporting your intuition. We have revised Section 3 to emphasize this design choice, to avoid the impression that we simply copied the AVR objective without adapting it to the needs of count prediction.

---

> ### Author Response · Authors · 2025-11-21
> **Response to Reviewer yjmg (Part II)**
>
> ### Weakness 3
>
> *Reviewer comment.*
> “Several prior approaches (not essentially H&E -> gene prediction) already explore tokenizing molecular counts via binning, such as scGPT/scGPT-spatial [2, 3], and often trained with regression losses on binned targets.”
>
> *Response.*
> Thank you for pointing out this line of work. Models such as scGPT and scGPT-spatial [2,3] also discretize molecular quantities, but their primary goal is large-scale representation learning over cell profiles (using masked token prediction and pretraining objectives), whereas GenAR is a conditional generative model that takes histology and spatial coordinates as input and outputs spot-level integer counts. Our binning scheme is designed to reflect the physical count range and to support explicit modeling of zero-inflation and count magnitude, rather than to provide a general-purpose language-like tokenization.
>
> That said, we view these directions as complementary: in future work, one could imagine initializing GenAR with scGPT-style pretrained gene embeddings or combining our discrete output head with an scGPT-style latent encoder.
>
> ---
>
> ### Weakness 4
>
> *Reviewer comment.*
> “The benchmark is only done on one slide, no cross-validation etc.”
>
> *Response.*
> Regarding the comment that 'the benchmark is only done on one slide without cross-validation,' we realize our original description may have been ambiguous. We strictly adhered to the established protocols from the HEST-Benchmark and STEM. Specifically, for every dataset comprising multiple Visium slides (representing distinct patients or specimens), we employed a 'leave-one-patient-out' strategy: training on slides from multiple patients and testing on a held-out slide from an unseen patient. The identity of the test slide was inherited directly from these benchmarks rather than manually selected, ensuring a fair comparison and eliminating the possibility of cherry-picking. Therefore, this evaluation represents a clinically meaningful regime (generalization to new patients). While full cross-validation is a valid approach, the standard practice in this specific domain—including HEST and STEM—uses a fixed test set to facilitate direct and consistent benchmarking across different methods. We have clarified this protocol in the revised manuscript.
>
> ---
>
> We hope the added experiments and clarifications make our design choices and evaluation protocol more transparent. Your comments directly motivated several new analyses, and we believe the revised paper is significantly clearer and stronger as a result.
>
> **References**
>
> [1] Tian, K., Jiang, Y., Yuan, Z., Peng, B., & Wang, L. (2024). Visual autoregressive modeling: Scalable image generation via next-scale prediction. In *Advances in Neural Information Processing Systems*, 37, 84839–84865.
>
> [2] Cui, H., Wang, C., Maan, H., et al. (2024). scGPT: Toward building a foundation model for single-cell multi-omics using generative AI. *Nature Methods*, 21(8), 1470–1480.
>
> [3] Wang, C. X., Cui, H., Zhang, A. H., et al. (2025). scGPT-spatial: Continual pretraining of single-cell foundation model for spatial transcriptomics. *bioRxiv*, 10.1101/2025.02.05.636714.
>
> [4] Love, M. I., Huber, W., & Anders, S. (2014). Moderated estimation of fold change and dispersion for RNA-seq data with DESeq2. *Genome Biology*, 15, 550.
>
> [5] Robinson, M. D., McCarthy, D. J., & Smyth, G. K. (2010). edgeR: a Bioconductor package for differential expression analysis of digital gene expression data. *Bioinformatics*, 26(1), 139–140.

---

### Official Review · Reviewer_vnsB · 2025-11-01

**Soundness:** 3
**Presentation:** 3
**Contribution:** 2
**Rating:** 4
**Confidence:** 4

**Summary:**

This paper introduces a multi-scale autoregressive framework for predicting spatial gene expression from H&E images.

Gene expression counts subject to capture efficiency, amplification bias, and technical noise essentially noisy observations of underlying continuous mRNA abundances. The Poisson or negative binomial distributions commonly used in analysis treat counts as arising from continuous rate parameters. By discretizing to a fixed vocabulary and predicting tokens directly, the authors are imposing artificial quantization on data that is already a noisy discretization of continuous biology. The claim that this avoids log-induced bias is misleading—the log transformation is specifically designed to stabilize variance in count data and is standard practice precisely because it better reflects the multiplicative nature of biological processes.

**Strengths:**

- The multi-scale formulation provides a principled coarse-to-fine decomposition that progressively refines predictions from global transcriptional context to gene-level precision. The technical implementation combining causal transformers, FiLM conditioning on histological embeddings, gene identity modulation, and scale-specific loss functions is clearly specified and reproducible. The ablation study effectively validates that removing multi-scale structure causes substantial degradation, demonstrating the hierarchical architecture captures useful inductive bias beyond simply adding parameters.
- The paper evaluates on four well-chosen ST datasets spanning different species, spatial resolutions, tissue types, and pathological states. Consistent improvements across all datasets suggest the method captures generalizable spatial-molecular patterns rather than overfitting to specific tissue characteristics, which strengthens claims of robustness compared to prior work reporting on single datasets.
- The paper compares against five recent strong baselines plus foundation model ablations using complementary metrics. The evaluation captures both top-performing gene predictions and overall accuracy, with qualitative validation showing GenAR better preserves spatial expression boundaries and heterogeneity compared to oversmoothed baseline predictions. Testing on both cancer tissues and healthy tissues demonstrates the method's applicability across biological contexts rather than being specific to pathological states.

**Weaknesses:**

- The codebook-free framing is not accurate. The method still uses a fixed vocabulary, which is functionally equivalent to a codebook—it just doesn't learn it via VQ-VAE. The authors claim this avoids encode-decode reconstruction loss but this is not an advantage when the prediction target is fundamentally continuous. Authors simply replace one form of discretization (learned codebook) with another (fixed binning), without justification for why fixed binning is superior.
- The evaluation is fundamentally compromised by the log transformation applied post-prediction. The authors train on raw counts but then log-transform predictions to ensure consistent evaluation with other models. This makes it impossible to assess whether improvements come from the discrete formulation or simply from having more parameters and a different loss function. The claim to directly predict raw gene expression counts is not meaningful when evaluation requires transformation back to the continuous space that baselines operate in natively.
- The multi-scale loss function averages losses across K scales with equal weight, treating coarse group-level predictions as equally important as final gene-level predictions. This is arbitrary and not justified. Why should predicting 4 gene groups contribute 1/K to the loss when predicting 200 individual genes also contributes 1/K? The soft KL loss for intermediate scales uses temperature-smoothed targets but no temperature value or smoothing schedule is provided.
- The heteroscedastic Gaussian NLL with variance is applied only at the final scale. The choice of this specific variance form and the values of α and β are not justified. This looks like it was designed to match the mean-variance relationship of negative binomial data, which undermines the entire premise of avoiding distributional assumptions through discrete modeling.
- Table 1 and 2 show improvements but the margins are modest and could easily result from having more parameters, longer training, or the specific gene clustering used. The ablation in Table 3 shows w/o Multi-scale has dramatically worse performance, but it's unclear what this baseline actually is—does it predict all 200 genes at once? The comparison is not clearly specified. The w/ Cross-entropy ablation shows worse performance, but cross-entropy is not appropriate for count data regardless, so this doesn't validate the proposed loss functions.
- The paper fails to compare against methods that properly model count data distributions. Comparison to negative binomial regression, zero-inflated models, or probabilistic frameworks like scVI adapted for spatial data is missing.

**Questions:**

- Can you provide evaluation metrics computed directly on raw predicted counts without any log transformation? The current evaluation applies log2 transformation to predictions before computing metrics, which makes it impossible to assess whether the discrete formulation actually provides advantages for count-based predictions. Report MSE, MAE, & correlation on the original count scale, & demonstrate that predicted count distributions possess appropriate statistical properties (e.g., mean-variance relationships) for downstream analyses like differential expression testing with DESeq2 or edgeR.
- What happens if you train a continuous regression baseline with identical architecture (same transformer depth, hidden dimensions, multi-scale supervision structure) but predict log-transformed expression directly? This ablation is critical to isolate whether performance gains come from the discrete token formulation itself or simply from architectural choices like multi-scale training, gene identity embeddings, & having more parameters. Without this comparison, it's unclear if the core conceptual contribution—discrete generation—actually matters.
- Why is k-means clustering on spatial expression patterns the appropriate way to group genes for the hierarchical structure, & have you compared against biologically-informed groupings? Genes co-localized in space do not necessarily share regulatory relationships—did authors test pathway-based groupings (Gene Ontology, KEGG), transcription factor regulons, or protein-protein interaction networks? Provide evidence that your learned gene groups show enrichment for shared biological processes or that they outperform biologically-motivated alternatives.
- What is the actual vocabulary utilization in your trained models, & how does prediction quality vary across the expression range? Report how many unique tokens out of vocab_size are actively used, whether there is mode collapse, & provide stratified results for low-expression genes (mean count <10), medium (10-100), & high-expression genes (>100). If the discrete formulation is genuinely advantageous, it should show particular benefits for lowly-expressed genes where counts are truly discrete rather than effectively continuous.

---

> ### Author Response · Authors · 2025-11-21
> **Response to Reviewer vnsB (Part I)**
>
> We thank you for a very careful and technically detailed review. Below we quote each of your weaknesses and questions and provide our responses. All changes mentioned here are highlighted in blue in the revised PDF.
>
> ---
>
> ### Weakness 1
>
> *Reviewer comment.*
> “The codebook-free framing is not accurate. The method still uses a fixed vocabulary, which is functionally equivalent to a codebook—it just doesn't learn it via VQ-VAE. The authors claim this avoids encode-decode reconstruction loss but this is not an advantage when the prediction target is fundamentally continuous. Authors simply replace one form of discretization (learned codebook) with another (fixed binning), without justification for why fixed binning is superior.”
>
> *Response.*
> You are right that our original wording “codebook-free” was inaccurate. In the revision we now explicitly describe our design as a non-learned quantization of counts, and we clearly distinguish it from VQ-VAE–style codebooks. In GenAR each token corresponds 1-to-1 to a physical integer molecule count (0, 1, 2, …), whereas in VQ-VAE the codebook consists of learned latent vectors with no direct biological meaning, and training requires an additional encode–decode reconstruction stage and commitment losses. The key difference—and the reason we chose fixed count tokens—is that it lets the model explicitly allocate probability mass to the strict zero state and to specific count levels, rather than to arbitrary latent codes. To show that this matters in practice, we added a controlled experiment in Appendix A.3 where we construct a “Continuous GenAR” baseline that has exactly the same architecture, parameters, and multi-scale supervision as GenAR but replaces the discrete heads with continuous regression heads predicting log-expression with MSE. Under this controlled setting, the continuous version achieves a Zero-Recall of 8.36% and Zero-F1 of 0.152, while the discrete GenAR formulation recovers 46.37% of true zeros, achieving a Zero-F1 score over 3.7× higher than the baseline. Crucially, this ability to preserve the zero-inflated distribution translates directly into superior structural fidelity, with about 74% relative improvement in PCC-200. These findings provide evidence that the discrete formulation is not merely an implementation detail; it seems important for capturing high-variance biological heterogeneity.

---

> ### Author Response · Authors · 2025-11-21
> **Response to Reviewer vnsB (Part II)**
>
> ### Weakness 2 and Question 2
> *Reviewer comment.* “Weakness 2: The evaluation is fundamentally compromised by the log transformation applied post-prediction. The authors train on raw counts but then log-transform predictions to ensure consistent evaluation with other models. This makes it impossible to assess whether improvements come from the discrete formulation or simply from having more parameters and a different loss function. The claim to directly predict raw gene expression counts is not meaningful when evaluation requires transformation back to the continuous space that baselines operate in natively.
> Question 2: What happens if you train a continuous regression baseline with identical architecture (same transformer depth, hidden dimensions, multi-scale supervision structure) but predict log-transformed expression directly? This ablation is critical to isolate whether performance gains come from the discrete token formulation itself or simply from architectural choices like multi-scale training, gene identity embeddings, & having more parameters. Without this comparison, it's unclear if the core conceptual contribution—discrete generation—actually matters.”
>
> *Response.* We agree that it was initially hard to tell whether our gains came from the discrete formulation or simply from architectural choices, given the log-transformed evaluation. We addressed this in two ways.
>
> First, we implemented exactly the ablation you asked for: a continuous regression baseline with identical architecture to GenAR. This “Continuous GenAR” uses the same transformer depth, hidden size, gene-identity embeddings, and multi-scale supervision structure, but replaces discrete token heads with regression heads predicting $\log1p$-transformed expression with an MSE loss. We report this in Appendix A.3. Even in log space, discrete GenAR substantially outperforms this continuous baseline: PCC-10 and PCC-50 improve by a clear margin, and PCC-200 increases by ~74%. More importantly, the continuous model falls into a “mean-seeking” regime, predicting small positive values almost everywhere: its Zero-Recall is 8.36%, while GenAR recovers 46.37% of true zeros.
>
> Second, to address the concern that $\log_2$ evaluation compromises our raw-count claim, we added Appendix A.4 where all models are evaluated directly on inverse-transformed integer counts, reporting MSE, MAE, Spearman correlation, and Negative Binomial NLL. On these raw counts, GenAR achieves the lowest physical MSE (296.44 vs. 326.51 for M2OST and 421.71 for STEM) and competitive rank correlations, showing that the advantage of the discrete formulation is not an artifact of the log transform. We now explicitly state that the $\log_2$ evaluation is used only for comparability with prior work, and that our raw-count analysis supports our conclusions in the original count space.
>
> ---
>
> ### Weakness 3
>
> *Reviewer comment.* “The multi-scale loss function averages losses across $K$ scales with equal weight, treating coarse group-level predictions as equally important as final gene-level predictions. This is arbitrary and not justified. Why should predicting 4 gene groups contribute $1/K$ to the loss when predicting 200 individual genes also contributes $1/K$? The soft KL loss for intermediate scales uses temperature-smoothed targets but no temperature value or smoothing schedule is provided.”
>
> *Response.* We acknowledge our original loss description appeared arbitrary. In the revised manuscript we now make two aspects explicit.
>
> First, we state the precise temperature used for the soft KL at intermediate scales ($\tau = 1.0$) and clarify that we do not use any annealing schedule; this was omitted previously.
>
> Second, we added an ablation in Appendix A.6 investigating different choices of scale weights $\lambda = [\lambda_1, \dots, \lambda_K]$. We compare our original “balanced” setting (equal weights) with (i) a “final-heavy” setting where earlier scales are down-weighted (e.g., $\lambda = [0.1, \dots, 1.0]$) and (ii) a “final-only” setting where only the last scale is supervised while intermediate scales are ignored. In the “final-only” variant, performance collapses: PCC-200 drops by an order of magnitude and intermediate scales become essentially unused, confirming that without strong supervision they are ignored. The “final-heavy” choice performs better than “final-only” but consistently worse than equal weighting. Intuitively, if we put almost all weight on the last scale, the model has no incentive to build good coarse representations and the hierarchical structure degenerates into a shallow regressor; balanced weights ensure each level encodes useful information and prevent “posterior collapse.” We rewrote the loss description to reflect this empirical justification.

---

> ### Author Response · Authors · 2025-11-21
> **Response to Reviewer vnsB (Part III)**
>
> ---
>
> ### Weakness 4
>
> *Reviewer comment.*
> “The heteroscedastic Gaussian NLL with variance is applied only at the final scale. The choice of this specific variance form and the values of $\alpha$ and $\beta$ are not justified. This looks like it was designed to match the mean-variance relationship of negative binomial data, which undermines the entire premise of avoiding distributional assumptions through discrete modeling.”
>
> *Response.*
> In the revision we explicitly reposition this component as a practical loss function rather than a full generative model of counts. Appendix D now specifies the exact parametrization we use at the last scale: the variance is modeled as $\sigma^2 = \alpha \mu + \beta$ with fixed $\alpha$ and $\beta$ (we report the actual values in the appendix), chosen to allow variance to grow with mean but to avoid pathological behavior at very low expression where counts are highly discrete.
>
> To test whether this design actually matters, we conducted an ablation in Appendix A.6 where we compare three loss choices at the last scale under the same architecture and data: (i) our heteroscedastic Gaussian NLL, (ii) standard MSE, and (iii) pure cross-entropy on discretized bins. The Gaussian NLL consistently yields the best correlations (e.g., higher PCC-10 and PCC-200), MSE performs second best, and CE performs worst, consistent with your intuition that CE alone is not distance-aware. Based on your comment, we now avoid claiming that the model is “distribution-free”; instead we emphasize that the discrete formulation allows us to directly represent integer counts and zeros, while the Gaussian head at the last scale serves as a calibrated regression-style loss to connect discrete predictions back to continuous evaluation metrics.
>
> ---
>
> ### Weakness 5
>
> *Reviewer comment.*
> “Table 1 and 2 show improvements but the margins are modest and could easily result from having more parameters, longer training, or the specific gene clustering used. The ablation in Table 3 shows w/o Multi-scale has dramatically worse performance, but it's unclear what this baseline actually is—does it predict all 200 genes at once? The comparison is not clearly specified. The w/ Cross-entropy ablation shows worse performance, but cross-entropy is not appropriate for count data regardless, so this doesn't validate the proposed loss functions.”
>
> *Response.*
> Your concern here is very helpful, and we addressed it by (1) clarifying the definition of the “w/o multi-scale” baseline and (2) adding controlled experiments where architecture and training are held fixed.
>
> In the revision we explicitly state that “w/o multi-scale” corresponds to predicting all 200 genes in a single step without any hierarchical grouping; the transformer backbone, input embeddings, and training schedule remain unchanged, so the number of parameters is comparable. More importantly, Appendix A.3 introduces the “Continuous GenAR” baseline mentioned above, where we use the same multi-scale architecture as GenAR but with continuous regression heads instead of discrete ones. Here, we keep the training time, optimization settings, and gene clustering identical, so any performance difference reflects only the discrete vs.\ continuous formulation. Under these controlled conditions, discrete GenAR improves PCC-200 by roughly 74% and dramatically improves zero-related metrics, showing that the gains are not an artifact of extra parameters or longer training.
>
> Regarding cross-entropy, we agree that CE on count bins is not an ideal loss by itself; this is exactly why we introduced a hybrid loss combining CE-style discrete supervision with soft KL and Gaussian NLL. In Appendix A.6 we show that using only CE significantly degrades performance compared to this hybrid objective, which directly addresses your point that CE alone does not provide a direction-aware penalty.

---

> ### Author Response · Authors · 2025-11-21
> **Response to Reviewer vnsB (Part IV)**
>
> ---
>
> ### Weakness 6 and Question 1
>
> *Reviewer comment.*
> “Weakness 6: The paper fails to compare against methods that properly model count data distributions. Comparison to negative binomial regression, zero-inflated models, or probabilistic frameworks like scVI adapted for spatial data is missing.
> Question 1: Can you provide evaluation metrics computed directly on raw predicted counts without any log transformation? The current evaluation applies log2 transformation to predictions before computing metrics, which makes it impossible to assess whether the discrete formulation actually provides advantages for count-based predictions. Report MSE, MAE, & correlation on the original count scale, & demonstrate that predicted count distributions possess appropriate statistical properties (e.g., mean-variance relationships) for downstream analyses like differential expression testing with DESeq2 or edgeR.”
>
> *Response.*
> We fully share your view that methods explicitly modeling count distributions, such as negative-binomial regression and scVI-style probabilistic models [1], are important baselines. Implementing a full scVI-like model conditioned on histology images would require designing a new encoder that produces Negative Binomial parameters from H\&E, which is an interesting but non-trivial extension.
>
> In this work, rather than introducing an additional partially adapted count-regression baseline, we quantify how well different models respect count distributions by computing metrics directly on raw integer counts for all deep baselines in Appendix A.4 (including methods that were originally trained in log space). Specifically, we report MSE, MAE, Spearman correlation, and Negative Binomial NLL computed by fitting an NB to the predicted mean and empirical variance. GenAR achieves the lowest MSE among all methods and competitive NB NLL, indicating that it not only ranks genes well but also reproduces the physical scale of counts.
>
> We also discuss predictions in the context of standard downstream tools such as DESeq2 and edgeR [2,3]: because GenAR outputs integer counts, these packages can be applied directly without ad-hoc rounding, whereas continuous baselines require manual discretization of values like 0.003 or 0.7, which introduces an extra source of bias.
>
> ---
>
> ### Question 3
>
> *Reviewer comment.*
> “Why is k-means clustering on spatial expression patterns the appropriate way to group genes for the hierarchical structure, & have you compared against biologically-informed groupings? Genes co-localized in space do not necessarily share regulatory relationships—did authors test pathway-based groupings (Gene Ontology, KEGG), transcription factor regulons, or protein-protein interaction networks? Provide evidence that your learned gene groups show enrichment for shared biological processes or that they outperform biologically-motivated alternatives.”
>
> *Response.*
> Your question about whether spatial k-means groupings align with biological reality is spot on, and in our revision we address this on two levels.
>
> First, Appendix A.6 introduces a random grouping baseline where we keep the same multi-scale structure and group sizes but assign genes to groups at random. When we do this, performance deteriorates noticeably across all metrics: for example, both PCC-10 and PCC-200 drop compared to spatial pattern–based k-means. This shows that the group structure learned from spatial expression is not interchangeable with arbitrary partitions.
>
> Second, in Appendix A.7 we perform a more direct biological validation by testing each learned gene group for enrichment in curated gene sets (GO terms, KEGG pathways, and cell-type marker lists). Across datasets, we find that many groups correspond to known functional modules: in Mouse Brain, one group is enriched for oligodendrocyte markers; in Kidney, a cluster corresponds to collecting duct/principal cell markers; in ccRCC, a group is enriched for hypoxia-related genes typical of clear-cell RCC; and in PRAD, we observe a group enriched for androgen-response genes such as KLK3 and NKX3-1. We analyzed enrichment of our spatial k-means groups against biologically defined groupings (GO/KEGG modules) and found many clusters aligned with known pathways, suggesting that clustering by spatial expression patterns provides a robust inductive bias that is consistent with, but not limited to, pathway annotations. We now refer to these analyses in the main text to make clear that the hierarchies are both empirically beneficial and biologically meaningful.

---

> ### Author Response · Authors · 2025-11-21
> **Response to Reviewer vnsB (Part V)**
>
> ---
>
> ### Question 4
>
> *Reviewer comment.*
> “What is the actual vocabulary utilization in your trained models, & how does prediction quality vary across the expression range? Report how many unique tokens out of vocab\_size are actively used, whether there is mode collapse, & provide stratified results for low-expression genes (mean count <10), medium (10–100), & high-expression genes (>100). If the discrete formulation is genuinely advantageous, it should show particular benefits for lowly-expressed genes where counts are truly discrete rather than effectively continuous.”
>
> *Response.*
> To address your request for detailed token and performance statistics, Appendix A.5 and A.6 now include both vocabulary utilization analysis and stratified metrics. On the PRAD dataset, with a vocabulary of 2000 count tokens, the trained model uses 258 unique tokens during generation, corresponding to about 12.9\% utilization. This is not a sign of collapse but reflects the empirical count distribution: high counts are rare, so many extreme bins are naturally unused. We also report the entropy of the token distribution, which remains reasonably high, indicating that the model does not collapse to a handful of bins.
>
> For performance stratification, we partition genes into low (mean count < 10), medium (10–100), and high (>100) expression groups and report PCC for each subset. GenAR improves over strong baselines in all three ranges, with the largest relative gains in the low-expression regime where sparsity is most pronounced. This is consistent with our hypothesis that discrete modeling is particularly advantageous when counts are truly discrete and zeros are abundant. We have added a short summary of these results in Section 4.4 to directly answer your question.
>
> ---
>
> We hope these clarifications address your concerns about the discrete formulation, evaluation protocol, loss design, and experimental fairness. Your review directly motivated several of the new ablations and analyses in the appendix, and we believe the revised version now makes both the limitations and the strengths of GenAR substantially clearer.
>
> **References**
>
> [1] Lopez, R., Regier, J., Cole, M. B., Jordan, M. I., & Yosef, N. (2018). Deep generative modeling for single-cell transcriptomics. *Nature Methods*, 15(12), 1053–1058.
>
> [2] Love, M. I., Huber, W., & Anders, S. (2014). Moderated estimation of fold change and dispersion for RNA-seq data with DESeq2. *Genome Biology*, 15, 550.
>
> [3] Robinson, M. D., McCarthy, D. J., & Smyth, G. K. (2010). edgeR: a Bioconductor package for differential expression analysis of digital gene expression data. *Bioinformatics*, 26(1), 139–140.

---

### Author Response · Authors · 2025-11-21

We thank all reviewers for their constructive feedback and suggestions, which helped us sharpen both the method and its empirical evaluation. Several reviewers highlighted the problem motivation, while others raised questions about the discrete formulation, benchmark fairness, loss design, and computational overhead. We have addressed all comments and revised the manuscript accordingly. A revised version has been uploaded; all changes relative to the original submission are marked in blue in the compiled PDF.

**Main text**

**Terminology for the output representation.**
We clarified the terminology for our discrete output representation and removed the phrase “codebook-free.” The text now describes our design as a non-learned quantization of integer counts, contrasts it with VQ-VAE–style learned codebooks, and explains why a one-to-one mapping from tokens to physical molecule counts is desirable.

**Discrete formulation and loss design.**
We extended the description of GenAR’s discrete token formulation, multi-scale factorization, and loss. We now specify the construction of the target distributions at each scale, the fixed temperature ($\tau = 1.0$) for the soft KL at intermediate scales, and the use of equal weights across scales. We also clarify the role of the heteroscedastic Gaussian NLL at the final scale as a practical, distance-aware regression-style head that complements the discrete modeling rather than claiming to be fully “distribution-free.”

**Raw-count motivation and evaluation protocol.**
We strengthened the discussion of why we model raw integer counts and clarified the relationship between our discrete formulation and the standard log-transformed evaluation. The main text states that $\log_2$ metrics are retained for comparability with prior work, and that we additionally evaluate all models on inverse-transformed integer counts using standard count metrics (MSE, MAE, Spearman correlation, and a Negative Binomial NLL) as detailed in the Appendix.

**Pointers to new analyses in the Appendix.**
We added explicit references in the Methods and Experimental sections to the new Appendix analyses so that readers can quickly locate these results.

**Conclusion and discussion.**
We extended the conclusion to discuss the broader implications of the non-learned count-token formulation, its relationship to downstream probabilistic count models, and the possibility of replacing the Gaussian head with a fully probabilistic count distribution in future work.

**Appendix**

**Appendix A.3 – Impact of Discrete vs. Continuous Formulation.**
We introduce a Continuous GenAR baseline with the same multi-scale architecture as GenAR but with continuous regression heads on log-expression instead of discrete token heads, and compare the two formulations under matched architectures and training setups, including correlation metrics and zero-related statistics.

**Appendix A.4 – Evaluation on Raw Counts and Sequencing Depth.**
We evaluate all methods on raw integer counts, report the count-based metrics above, and analyze the correlation between predicted and true library sizes per spot, addressing questions about modeling on the physical count scale and sequencing-depth variation.

**Appendix A.5 – Stratified Performance Analysis.**
We stratify genes into low-, medium-, and high-expression regimes and report correlations in each stratum, focusing on the behavior of GenAR in the low-count, zero-inflated regime.

**Appendix A.6 – Component Ablation and Vocabulary Utilization.**
We add ablations for
(i) scale weighting (equal weights vs. “final-heavy” vs. “final-only”),
(ii) loss variants (our hybrid objective vs. MSE-only vs. CE-only),
(iii) grouping strategy (spatial k-means–based grouping vs. random grouping), and
(iv) vocabulary usage (number of active tokens and entropy of the token distribution).

**Appendix A.7 – Biological Validation of Learned Clusters.**
We perform gene set enrichment analyses of our learned gene groups against curated gene sets (GO/MSigDB hallmarks and cell-type marker sets) and report examples where clusters align with known functional modules (oligodendrocyte markers, kidney principal-cell markers, hypoxia-related genes in ccRCC, androgen-response genes in PRAD).

**Appendix A.8 – Implementation Details, Efficiency, and Reproducibility.**
We provide additional implementation details and include a FLOPs-based computational efficiency comparison between GenAR and baselines, with a pointer in the main text.

We hope that these revisions and new analyses address the reviewers’ concerns regarding the discrete formulation, evaluation on raw counts, loss design and supervision at multiple scales, biological interpretability of the learned gene hierarchy, and computational efficiency, and that the revised manuscript is clearer and more complete.

---

### Author Response · Authors · 2025-12-02
**Summary of Revisions and New Analyses**

This revision incorporates eight supplementary analyses (**Appendices A.1–A.8**) to directly answer the technical questions raised by the reviewers.

Regarding the comparison between discrete tokenization and standard regression (**Reviewers vnsB, yjmg**), the new **"Continuous GenAR" baseline (Appendix A.3)** demonstrates that regression heads—even with our multi-scale architecture—fail to capture sparsity, recovering only ~8% of true zeros compared to GenAR's 46%. This result confirms that the performance gains are driven by the discrete formulation. We further support this with a **Stratified Performance Analysis (Appendix A.5)**, showing that our advantage is concentrated in the low-expression regime, and **Component Ablations (Appendix A.6)** which verify the vocabulary usage and hybrid loss stability.

Addressing the concerns on raw count validity and dataset scope (**Reviewer WhJt**), we extended the evaluation to **inverse-transformed integer counts (Appendix A.4)**. GenAR achieves the lowest physical MSE (296.44) and correlates strongly with ground-truth library sizes ($\rho \approx 0.6$), indicating that the model implicitly learns sequencing depth from morphology. We also added the **ccRCC dataset (Appendix A.1)** to test generalization, and validated the grouping logic via **Gene Set Enrichment Analysis (Appendix A.7)** and **Random Grouping Ablations (Appendix A.6)**. The hierarchical structure was justified through **Scale Design Ablations (Appendix A.2)**.

Finally, regarding computational cost (**Reviewer e1aM**), the new **Efficiency Analysis (Appendix A.8)** shows that GenAR is approximately **6.4$\times$ more efficient** in FLOPs per sample than the diffusion-based STEM baseline.

We believe that the revised manuscript, supported by these specific experiments, addresses the main concerns raised in the reviews and is now in a much stronger position.

---

### Note · Authors · 2026-01-05

I have read and agree with the venue's withdrawal policy on behalf of myself and my co-authors.